# The Cellular Response Is Determined by a Combination of Different ELF-EMF Exposure Parameters: A Scope Review

**DOI:** 10.3390/ijms25105074

**Published:** 2024-05-07

**Authors:** Isabel López de Mingo, Marco-Xavier Rivera González, Ceferino Maestú Unturbe

**Affiliations:** 1Centro de Tecnología Biomédica (CTB), Universidad Politécnica de Madrid (UPM), 28223 Madrid, Spain; isabel.lopez@ctb.upm.es (I.L.d.M.); marco.rivera@ctb.upm.es (M.-X.R.G.); 2Escuela Técnica Superior de Ingenieros de Telecomunicación (ETSIT), Universidad Politécnica de Madrid (UPM), 28040 Madrid, Spain; 3Escuela Técnica Superior de Ingenieros Informáticos (ETSIINF), Universidad Politécnica de Madrid (UPM), 28223 Madrid, Spain; 4Centro de Investigación en Red—Bioingeniería, Biomateriales y Nanomedicina (CIBER-BBN), 28029 Madrid, Spain

**Keywords:** ELF-EMF, cells, frequency, intensity, magnetic fields

## Abstract

Since the establishment of regulations for exposure to extremely low-frequency (0–300) Hz electromagnetic fields, scientific opinion has prioritised the hypothesis that the most important parameter determining cellular behaviour has been intensity, ignoring the other exposure parameters (frequency, time, mode, waveform). This has been reflected in the methodologies of the in vitro articles published and the reviews in which they are included. A scope review was carried out, grouping a total of 79 articles that met the proposed inclusion criteria and studying the effects of the different experiments on viability, proliferation, apoptosis, oxidative stress and the cell cycle. These results have been divided and classified by frequency, intensity, exposure time and exposure mode (continuous/intermittent). The results obtained for each of the processes according to the exposure parameter used are shown graphically to highlight the importance of a good methodology in experimental development and the search for mechanisms of action that explain the experimental results, considering not only the criterion of intensity. The consequence of this is a more than necessary revision of current exposure protection regulations for the general population based on the reductionist criterion of intensity.

## 1. Introduction

### 1.1. Introduction Rationale

The study of interactions between magnetic fields and biological systems has been of great scientific interest since the 1980s [1,2,3,4,5,6,7,8,9,10]. From that time until the establishment of exposure guidelines for low-frequency electromagnetic fields (ELF-EMFs), numerous cellular assays were performed that sought significance for alterations found in biological processes [11] such as intracellular calcium flux [1,2,3], mRNA transcription [4,5] or DNA synthesis [6].

Following the establishment of standards for exposure to time-varying electric, magnetic and electromagnetic fields (up to 300 GHz) based primarily on the energy dose received by the cell as a stationary entity [12], there have been numerous reviews published that attempt to compile the number of results that exist in in vitro studies in bioelectromagnetics [13,14,15,16,17,18]. These reviews have traditionally been related to the search for an affirmative or negative answer to the occurrence of certain alterations in cellular processes such as the appearance of tumours [14,19,20,21,22,23,24,25,26,27] or alterations in the immune system [13,28,29,30,31,32,33,34,35] and the reproductive system [36]. With some exceptions, the experimental results have been grouped together without establishing a division by exposure parameters (intensity, frequency, time, waveform, mode) used in the configuration of the exposure system, prioritising the discretisation of the result of the cellular process studied over the exhaustive control of the experimental conditions.

The fact that different combinations of exposure parameters are used makes it difficult to draw conclusions about the effects that these magnetic fields might produce in the different cell models studied [11,21,37,38,39,40,41]; there are many publications that detect non-homogeneity in the choice of parameters as the main reason for the non-replicability of the experiments and that claim that the results are inconclusive and incomparable among the existing publications [11,24,32,42,43,44,45].

This large gap in the literature on in vitro studies in bioelectromagnetics prevents researchers from determining the importance of the different exposure parameters in their experimental developments and, therefore, the search for mechanisms of action that may help to elucidate the results found in their research. Many hypotheses have been put forward [26,46,47,48,49,50,51,52,53,54], but few conclusions have been drawn from the research articles.

Extensively studied cellular processes have been cell viability and proliferation [31,55,56,57,58], apoptosis [16,31,44,59,60], oxidative stress and mitochondria [17,18,36,44,57,58,59,61,62,63], the cell cycle [31,56,59], cell signalling pathways such as the calcium-mediated pathway [15,30,58,64], protein alteration [57,59,63] and genetic effects [23,24,25,31,38,42,43,63], as target cellular processes both to try to apply magnetic field exposures as a possible clinical treatment in different clinical areas [37,50] including wound regeneration [45,65,66], bone repair [67,68,69,70,71], cancer [25] and ischaemic cerebral infarction [72].

This review aims to group articles on in vitro experiments in bioelectromagnetics based on the exposure parameters used (frequency, intensity, exposure time, exposure modality: continuous/intermittent) in the evaluation of the most studied cellular processes (viability, proliferation, apoptosis, oxidative stress and mitochondria, cell cycle), trying to identify differences in cell behaviour based on the parameters used.

### 1.2. Objective

The objective of this review is to analyse the cellular behavioural response of important cellular processes such as viability, proliferation, apoptosis, oxidative stress/mitochondria and the cell cycle when different cell lines are exposed to a magnetic field identifying the importance in the choice of exposure parameters (the frequency, intensity, time and continuous/intermittent mode of exposure). A bibliographic search was carried out in the NCBI PubMed and Scopus databases in which articles were identified that met a series of previously defined characteristics in order to narrow down the selection of items and to adjust the review to the main objective of the discussion, the importance of exposure parameters in the behavioural response of cells. The objectives of the review, inclusion/exclusion criteria and study design were defined by the PECOS guidelines (Population, Exposure, Comparison, Outcome, Study design). The PECOS statement was “What is the association between magnetic field exposure parameters and cellular response?”. As a consequence, the different sections of the PECO are as follows: in vitro models of immortalised or primary human or mouse/rat tumour and non-tumour cells (P); exposure to magnetic fields with frequencies (0–300] Hz with a continuous or intermittent exposure mode, intensities less than 1 T and any exposure time and waveform (E); sham-exposed (sham) control sample (P); alterations in viability, apoptosis, proliferation, oxidative stress levels and the cell cycle considering all the biomarkers (O) of molecular biology assays specific to the processes described above and original research in comparison with controls (no exposure) (S).

## 2. Methods

The scope review follows the PRISMA-ScR guidelines (Preferred Reporting Items for Systematic reviews and Meta-Analyses extension for Scoping Reviews) [73]. The methodology of this scoping review is registered in the Open Science Framework repository (https://osf.io/yp295) on 10 April 2024. The checklist is included in the Appendix A.

### 2.1. Eligibility Criteria

The selected articles had to be published in peer-reviewed scientific journals and be original research, so conference proceedings, reviews and book chapters were not included. The articles had to be written in English and be published in Open Access or be accessible through agreements between the scientific institution and the publisher.

We searched for studies performed on cell cultures in any cell line, without discarding articles for being immortalised lines/primary cell cultures, tumour/non-tumour, adherent/in suspension, human/animal cells or restricting the search to a specific tissue of origin. Only monolayer results have been considered in the articles studying 3D cultures or conventional monolayer cultures. The cell parameters chosen were viability, proliferation, apoptosis, oxidative stress and mitochondria injury and the cell cycle regardless of the biomarker used. Articles that did not study these cell parameters were discarded, e.g., genetic studies.

The exposure restrictions were magnetic field, with frequencies (0–300] Hz and intensities regardless of magnitude. No discrimination was made according to the exposure system used, neither by exposure time nor by continuous/intermittent mode. Original research articles that met all cellular requirements but did not show complete technical information related to magnetic field exposure (e.g., did not include frequency, intensity, exposure time, etc.) were discarded.

All articles comparing two distinct populations, one exposed to magnetic fields and one not exposed to the magnetic field (control) shielded or not by high permeability material (e.g., mu metal chamber), were included.

### 2.2. Information Sources and Search Strategy

The databases reviewed were NCBI PubMed and Scopus. Articles were selected from the year 2000 to 2023. The searches were performed on 4–6 November 2023. Both databases were queried as follows: “ELF-EMF AND cell”. In the case of NCBI PubMed, a total of 251 results were returned, 332 in the case of Scopus.

### 2.3. Selection of Sources of Evidence

All articles were selected based on the title and the abstract describing them. Those that did not meet the requirements established in the PECOS statement were excluded. They were selected independently between databases and then proceeded to the elimination of duplicates. The selected articles were imported in their complete version into the reference management software Mendeley^®^ (Mendeley Reference Manager, v. 2.110.0, Elsevier, London, UK). Once analysed in depth, we excluded those that did not fully meet the inclusion requirements. For all excluded items, the reason for exclusion is indicated (see Appendix A). The results of this filtering process can be seen in the PRISMA flow chart (Figure 1).

### 2.4. Data Charting Process and Data Items

Each of the selected articles has been carefully read, and the following information has been extracted in a summary table that has served as the basis for writing each of the sections of this review:Author and year of publication (full reference of the selected article);Cell line used;Frequency;Intensity;Exposure time;Exposure mode (intermittent/continuous);Results obtained: significant or non-significant for each cellular process studied (viability, proliferation, apoptosis, oxidative stress/mitochondria, cell cycle).

### 2.5. Synthesis of Results

Different divisions have been performed for the synthesis and analysis of the results:First division: continuous (C)/intermittent (I) exposure. The articles were separated according to whether the exposure was continuous over time or intermittent (ON/OFF) regardless of the exposure time.Second division: tumour cells (T)/non-tumour cells (NT). The results of the different articles are divided according to whether they use tumour or non-tumour cells independently of the tissue from which they originate.Third division: frequency. Two frequency divisions are established: exposures at frequencies less than or equal to 50 Hz and exposures greater than 50 Hz.Fourth division: intensity. Four levels of intensity are established: exposures at intensities less than or equal to 100 µT, exposures greater than 100 µT and less than or equal to 1 mT, exposures greater than 1 mT and less than or equal to 5 mT and, finally, exposures greater than 5 mT.Fifth division: time. Three exposure time bands are established: exposures less than or equal to 1 h; exposures greater than 1 h and less than or equal to 1 day; and, finally, exposures greater than 1 day.

The results are categorised as follows: non-significant (0), statistically significant decrease (1), statistically significant increase (2).

## 3. Results

### 3.1. Selection of Sources of Evidence

The results of the bibliographic search can be seen in Figure 2A, which shows the diagram of the identification of databases and records using the PRISMA-ScR protocol. As previously established, the PubMed and Scopus databases were used for item selection. A total of 583 articles were identified (PubMed *n* = 251; Scopus *n* = 332); of these, 222 duplicates, 23 that were not in English or not available for reading and 93 that were not performed in in vitro models were eliminated. A total of 245 articles were obtained, of which 150 were excluded for not meeting the requirements imposed: review/opinion article/book chapter (41), not measuring the selected cell parameters of viability, proliferation, apoptosis, oxidative stress, cell cycle (56), RF or static field (3), prior to the year 2000 (13), electric field (1) and in models other than in vitro (36). After these exclusions, the number of articles was reduced to 95. In the last screening step, 5 review/opinion article/book chapters and 11 articles that meet the cellular processes to be reported were excluded. The total number of items included in this review is 79. Figure 2B shows the articles finally excluded and included in this review according to the criteria imposed and the articles excluded according to the exclusion criterion (duplicate articles are not represented). As can be seen, the main reason for exclusion is not being performed on an in vitro model (45.74%) followed by not complying with the cellular processes to be studied (23.75%).

All articles considered in the database with exclusion criteria can be found in the Appendix A.

### 3.2. Characteristics of Sources of Evidence

Figure 3 shows the temporal distribution by the year of publication of the articles included in this review. Most of these articles were published in 2014–2015. An increase in valid articles is observed from 2012 onwards, with minimal articles included before 2010.

The different experiments carried out in the articles selected for the review were extracted, a total of 531 (Figure 4A). These experiments were divided according to specific exposure parameters: frequency, intensity, exposure time and exposure mode (continuous vs. intermittent). For each of them, the impact of the parameter on certain cellular processes was determined: viability (continuous: Table 1; intermittent: Table 2), proliferation (continuous: Table 3; intermittent: Table 2), apoptosis (continuous: Table 4; intermittent: Table 2), oxidative stress/mitochondria (continuous: Table 5; intermittent: Table 2) and cell cycle (continuous: Table 6; intermittent: Table 2). A further division was made according to experiments carried out on tumour vs. non-tumour cells.

Most of the experiments collected show results for viability processes (49.72%) followed by proliferation processes (15.82%), apoptosis (12.62%) and oxidative stress/mitochondria (12.05%) (Figure 4A). The process with the least results is the cell cycle (9.79%). Most of the experiments were performed on tumour cells (56.12%), with the cellular processes of apoptosis (T: 51; NT: 16) and the cell cycle (T: 18; NT: 34) showing the greatest decompensation in the use of tumour or non-tumour cells (Figure 4B).

### 3.3. The Results and Critical Appraisal of the Source of Evidence

Figure 5 shows the distribution of the experiment results over 100% of the set of experiments for a specific cellular process. The majority of the in vitro studies of magnetic field exposure (0–100] Hz using tumoral cells find a decrease in viability (30.30%), with a minimal proportion of articles finding an increase in viability (2.65%) (Figure 5A). In the case of non-tumour cells, most results show an increase in cell viability (23.86%), with the percentage of results showing a decrease in this process being almost nil (0.38%) (Figure 5A). In the case of the analysis of proliferation, when tumour cells are exposed to magnetic fields, most show a decrease in proliferation (25.00%), while in non-tumour cells, the percentage showing a decrease is lower (8.33%) than that showing an increase (14.29%) (Figure 5B). In the case of apoptosis, the same number of experiments with tumour cells show an increase (7.04%) as a decrease (7.04%) (Figure 5C). The apoptosis results in non-tumour cells show only increased apoptosis, although the percentage over the total is small (9.86%) (Figure 5C). The large percentage of non-significant results in the assessment of apoptosis in non-tumour cells (63.38%) is remarkable. The oxidative stress results in tumour cells show a higher proportion of cells responding with increased (32.81%) versus decreased oxidative stress (4.69%), which coincides with the proportion of non-tumour cells responding with increased (25.00%) versus decreased (4.69%) (Figure 5D). Finally, in the case of the cell cycle, the percentages of results showing statistically significant results in tumour cells (15.38%) and non-tumour cells (15.38%) are similar in proportion over the total (Figure 5E).

As shown in Figure 6, most experiments were performed in a frequency range of (0–50] Hz (80.04%), at intensities between (0.1–1] mT (31.83%) and (1–5] mT (32.2%), an exposure time between 1 and 24 h (44.63%) and in continuous exposure mode (85.23%).

#### 3.3.1. Frequency

Figure 7 shows the viability results obtained in the first division of a frequency range of (0–50] Hz and (50–100] Hz (Figure 7A,B).

Viability processes in tumour cells are studied in a higher proportion in frequency ranges up to 50 Hz (73.65%) (Figure 7A). They generally show a decrease in cell viability in both frequency groups (up to 50 Hz: 42.57%; above 50 Hz: 14.66%), and only 4.73% of studies in tumour cells show an increase in viability in the (0–50] Hz range (Figure 7A). In the case of non-tumour cells, the generalised response is an increase in viability for both frequency bands (Figure 7B). Again, the most studied frequencies are in the (0–50] Hz range (Figure 7B). No data of decreased viability are shown for this cell group (Figure 7B).

A second frequency division is established for the viability assays in the 20 Hz range from 0 to 100 Hz. Most studies are concentrated in the frequency range (40–60] Hz, and this frequency has the highest percentage of negative results (25.68%) (Figure 7C). In the case of non-tumour cells, the same frequency range is the most studied, also being the one that returns the highest viability values (38.79%) (Figure 7D).

In the case of proliferation, lower frequencies (0–50] Hz produce a greater decrease in the proliferative capacity of tumour cells (47.73%) compared to non-tumour cells that show a tendency to increase this capacity (30.00%) (Figure 8A,B). There are no results for tumour cells for frequencies higher than 50 Hz among the chosen experiments, but there are results for non-tumour cells with mostly non-significant results (17.50%).

The apoptosis assays show a very significant increase in non-significant results for the range of frequencies studied, both in the case of tumour cells and non-tumour cells (Figure 8C,D).

In the oxidative stress results, a considerable increase is observed in tumour cells in both frequency groups (up to 50 Hz: 40.54%; above 50 Hz: 16.22%) (Figure 8E). In non-tumour cells, there is an increase in oxidative stress at frequencies below 50 Hz (55.56%) (Figure 8F). At higher frequencies, we found a greater number of decreasing results (7.41), as opposed to increasing (3.70%) (Figure 8F).

Regarding the cell cycle, tumour cells show effects occurring at low frequencies (0–50] Hz (44.44%), with no effects at frequencies higher than 50 Hz (Figure 8G). In non-tumour cells, a higher percentage of effects is found at low frequencies (0–50] Hz (17.65%) with a lower percentage at frequencies higher than 50 Hz (5.88%) (Figure 8H). However, most results in non-tumour cells for both frequency ranges are non-significant (frequencies below 50 Hz: 26.47%; frequencies above 50 Hz: 50%) (Figure 8H).

#### 3.3.2. Intensity

Figure 9 shows the results in viability (Figure 9A,B), proliferation (Figure 9C,D), apoptosis (Figure 9E,F), oxidative stress (Figure 9G,H) and cell cycle processes (Figure 9I,J) obtained according to intensity bands: (0–0.1] mT, (0.1–1] mT, (1–5] mT and higher than 5 mT.

The viability results show that in the case of tumour cells, high intensities show the highest percentage of decrease, (1–5] mT (31.08%) and higher than 5 mT (16.89%) (Figure 9A). In the case of non-tumour cells, viability increase results are shown in the range of (1–5] mT (42.24%), being the percentages similar in results that have no significant effect (6.90%) to those that obtain an increase in viability (7.76%) at intensities higher than 5 mT (Figure 9B).

In the case of proliferation, tumour cells show increased results of a decreased proliferative capacity at intensities of (0.1–1] mT (34.09%) (Figure 9C). At lower intensities (0–0.1] mT, the results between increase (11.36%) and decrease (9.09%) are very similar (Figure 9C). In the case of non-tumour cells, significant results of an increased proliferative capacity are found at intensities of (0.1–1] mT (30.00%), although there is a high percentage of results showing non-significant results (37.50%) (Figure 9D). Also, in the same range of intensities, there is a decrease in this capacity, although in a much lower percentage (10.00%) (Figure 9D). The rest of the intensities show no significant results (Figure 9D).

In the apoptosis process, tumour and non-tumour cells show mostly non-significant results (Figure 9E,F). In non-tumour cells, an increase in cell death is shown at intensities of (0.1–1] mT (25.00%), but for all other intensities, the non-significant results are greater than the increase or decrease effects (Figure 9F).

Oxidative stress is increased in tumour cells in intensity ranges of (0–0.1] mT (13.51%), (0.1–1] mT (21.62%) and intensities higher than 5 mT (18.92%) (Figure 9G). In this case, non-tumour cells respond similarly in the same intensity ranges, except at intensities higher than 5 mT where the majority of the effects are decreasing (7.41%) (Figure 9H).

The cell cycle shows the highest number of effects at intensities of (0.1–1] mT in tumour (33.33%) and non-tumour (11.76%) cells (Figure 9I). However, a very high number of non-significant results are found for each of the intensity ranges (Figure 9J).

#### 3.3.3. Exposure Time

Figure 10A shows the results in viability (Figure 10A,B), proliferation (Figure 10C,D), apoptosis (Figure 10E,F), oxidative stress (Figure 10G,H) and cell cycle processes obtained according to exposure time bands: (0–1] h, (1–24] and more than 24 h.

Tumour cells show a similar trend in all the time slots collected, although there is a small increase in viability at exposure times longer than 24 h (4.73%) that does not appear at all other times (Figure 10A). In the case of non-tumour cells, the proportion of significant results decreases with increasing exposure time, (0–1] h (0: 3.45%; 1: 17.24%), (1–24] h (0: 21.55%; 1: 33.62%) and times longer than 24 h (0: 19.83%, 1: 3.45%) (Figure 10B).

In proliferation, tumour cells show a tendency to decrease in proliferative capacity as exposure time increases, (1–24] h (6.82%), times greater than 24 h (38.64%) (Figure 10C). In non-tumour cells, the trend is towards increased proliferation in all three time bands, although decreasing results begin to appear in cells exposed at times greater than 24 h (17.50%) (Figure 10D).

In the case of apoptosis, no differences are shown in tumour cells for any of the time bands studied (Figure 10E). In non-tumour cells, the proportion of increased apoptosis responses and non-significant results is very similar in the time bands where data are shown, (1–24] h (0: 43.75%; 2: 25.00%), times greater than 24 h (0: 12.50%; 2: 18.75%) (Figure 10F).

Oxidative stress processes show a proportional increase with exposure time in tumour cells, (0–1] h (8.11%), (1–24] h (16.22%) and times longer than 24 h (32.43%) (Figure 10G). In the case of non-tumour cells, the highest proportion of the results clustered around exposure times of (0–1] h (18.52%) and (1–24] h (33.33%), although in this case, this does not appear to be related to time course (Figure 10H).

Finally, the cell cycle results show an increase in effects with increasing exposure time in tumour cells (1–24] h (5.56%), times longer than 24 h (38.89%) (Figure 10I). In the case of non-tumour cells, there is no increase in the results as exposure time increases, and the proportion of non-significant results and the appearance of effects is proportional between the time slots studied (Figure 10J).

#### 3.3.4. Exposure Mode

Figure 11 shows the results in viability (Figure 11A), proliferation (Figure 11B), apoptosis (Figure 11C), oxidative stress (Figure 11D) and cell cycle (Figure 11E) processes obtained according to exposure mode, continuous (C) or intermittent (I).

In the case of viability assays, both continuous and intermittent modes of exposure show similar results, with intermittent exposure showing a slightly higher percentage of studies returning a decrease in viability (C: 30.68%; I: 38.10%) (Figure 11A). The proliferation results show an increase in the proliferative capacity of cells exposed to an intermittent mode of exposure (41.18%), compared to a continuous mode (22.62%) (Figure 11B). In the case of apoptosis assays, the same proportion of increased programmed cell death is shown in both continuous and intermittent exposure modes (C: 17.91%; I: 20.00%), with a very high proportion of non-significant results in both modes (C: 80.60%; I: 80.00%) (Figure 11C). Cells exposed to intermittent magnetic fields (82.35%) show an increase in oxidative stress compared to cells subjected to continuous exposure (57.81%) (Figure 11D). In the case of the cell cycle, the proportions of the occurrence of effects are similar for continuous exposures (30.77%) and intermittent exposures (40.00%) (Figure 11E).

## 4. Discussion

### 4.1. Summary of Evidence

The main objective of this review is to highlight the importance of the control of exposure parameters in bioelectromagnetic cellular assays. Each of the parameters studied and included in this review of the scientific literature (frequency, intensity, exposure time, exposure mode) determines the cellular response to certain cellular processes such as those described here (viability, proliferation, apoptosis, oxidative stress and mitochondrial damage and cell cycle). Consequently, the question arises as to which parameter is the most decisive in the cellular response to each of these processes. Although we have observed in the results of this review that there does not seem to be an obvious answer to this question, it is not clear which is the most important one.

For decades, due to the establishment of exposure regulations for low-frequency electromagnetic fields, intensity has been thought to be the determining parameter for most of the effects found in both in vitro and in vivo experiments [139,140,141,142]. The so-called “dose effect”, whereby the probability of effects increases proportionally to the intensity, has been one of the most reinforced theories in the scientific literature [139,140,141,142]. However, biological systems are not linear systems [47,143,144], which leads us to believe that even if the stimulus increases, the response will not. The results of this review show inconclusive results of the occurrence of effects by intensity ranges and therefore do not seem to conform to a dose effect correlation between this parameter and the response to the cellular processes studied (Figure 9).

Frequency has been studied by numerous researchers as a possible cause of resonance effects in the cell that could determine the responses found in in vitro studies [10,53,54,145,146,147,148,149,150,151,152]. Although most studies continue to be performed in the frequency range [50–60] Hz, as this is the range determined for the electricity distribution network in Europe and America, results are obtained for other frequency ranges, as can be seen in Figure 7. Some cell parameters appear to be more sensitive to frequency such as proliferation (Figure 8B), oxidative stress (Figure 8F) and the cell cycle (Figure 8G) in which changes in cell behaviour in different frequency ranges are evident. In turn, time, exposure mode and waveform (which has been left out of this review) seem to be able to determine the behaviour of biological systems. If we look at Figure 10, we can see that a longer duration of exposure does not always lead to a greater occurrence of results; we are, therefore, faced with a new non-linear response of biological systems; an increase in exposure time is not synonymous with an increase in response.

Perhaps the solution is not to think of one parameter as the conductor of the orchestra but rather a combination of these parameters that is specific to each cell type. We would understand this combination of parameters as a code that the cell receives and that sets in motion the different mechanisms of action. Therefore, in view of the results obtained, we reiterate the importance that must be given to the search for the effects that each of these parameters and their different combinations can produce in different cellular strains. One of the main problems in the implementation of cellular experiments of exposure to magnetic fields is that there is no objective quantification of the different parameters, being that the measurement of the frequency, intensity and real waveform that the cell receives on the culture surface is imprecise and inaccurate. This leads to contradictory results between different research groups that use the same cell line, the same intensity and frequency, and even the same waveform, but fail to mimic the methodologies for monitoring these parameters and maintain methodological rigour. For this purpose, a gaussmeter with a triaxial probe must be used that allows for the measurement of the magnetic field in the three spatial components (X,Y,Z) individually and not as a resultant vector. For the characterisation of the magnetic field signals, the gaussmeter must be able to sample the signal in real time. The Fourier transform of the recorded signal should be indicated in the methodology, highlighting the fundamental frequency of the indicated waveform, an analysis of the harmonics of the recorded signal and the type of the intensity measurement peak–peak value or root mean square (RMS) value. This allows for the replicability of the experiments and their reliability in the methodology. In addition, it is important to characterise and indicate in the methodology the intensity values recorded over the whole surface of the culture plate, in order to record inhomogeneities on the biological material surface.

When an attempt is made to group these scientific articles in literature reviews that seek to draw conclusions related to a particular biological process, the main discussion is about the inability to obtain conclusive results and the lack of scientific evidence. In these reviews, generally, the parameters are analysed together without discriminating by frequency, intensity, exposure time, waveform, exposure mode or cell line. These comparisons are not useful for obtaining repeat patterns nor do they feed the search for possible models of interaction between magnetic fields and biological systems. If the exposure parameters are not the same, the comparison is not feasible. This lack of consensus on the exposure parameters of magnetic fields is not the only thing that determines the non-replicability of experiments; the use of different biomarkers and cell types is also a determining factor [39,40]. In this review, a division of results by tumour and non-tumour cell type has been made. It has been decided to separate into these broad groups to support the hypothesis that magnetic fields could have a reproductive response by decreasing or increasing biological processes such as proliferation that determine tumour development. In the results presented, it appears that the type of cellular response may indeed be determined by the nature of the cell (tumour, non-tumour) (Figure 5), with cellular processes such as viability and proliferation returning opposite results depending on the cell type tested (Figure 5A,B). However, it is likely that each cell type shows a different response that cannot be unified into a single combination of parameters depending on whether the cell is tumour or non-tumour or belongs to one tissue or another.

Another aspect that is not often reported in the articles is the values of the background field that exists in the samples at the time of exposure. This value also includes the value of the geomagnetic field of the place where the exposure is taking place. In most articles, these values are not reported, so it is impossible to know the pre-exposure conditions in terms of the magnetic field of the room where the incubators are located and inside the incubator itself. These values have also been shown to influence cell behaviour and therefore the results obtained [39,40,41,153].

So far, the level of statistical significance used in the analysis of experimental data has been considered to support the occurrence of significant cellular effects, but no optimal underlying mechanism is available to explain statistical significance. Some of the proposed mechanisms cite transient radical pairs and triplet phenomena [154,155,156] or resonant effects [10,145,146,147,148,149,150,151,152]. The main theory of the mechanisms of action was based on the induction of electric fields and currents in biological tissues, but many of these phenomena require much lower energy levels at which the occurrence of these effects is limited, which could be explained by the existence of non-linear states. No single interaction model has been determined to cause the effects found, probably due to the interaction of more than one mechanism of action causing the effects. Determining the cellular response to a given combination of exposure parameters is not as simple as applying statistical models and concluding with significant or non-significant effects.

The existence of effects in in vitro assays on different cellular processes studied is a fact that cannot be disregarded because of the inability to draw conclusions in a field as heterogeneous as bioelectromagnetism. What is essential and necessary is the establishment of laboratory methodological rules that ensure replicability, objectivity (understood as the interpretation of results based on experimentation) and methodological rigour in the control of exposure parameters and cell analysis.

### 4.2. Limitations

This review has been based on the articles contained in only two databases (PubMed and Scopus), which may result in the loss of articles that are not contained in them. Although the search sequence was broad so as not to miss too much of the relevant scientific literature that met the characteristics of our inclusion criteria, it is possible that important references may have been lost due to the use of the two databases chosen.

Cell grouping for the determination of the effects has been performed on the bulk of two cell populations, tumour (T) cells and non-tumour (NT) cells; however, it would be logical and accurate to study each of these effects for each of the possible differences between lines, to begin with, the division between human cells and animal cells, also whether they are primary or immortalised cell lines, following with suspension or adherent cell lines’ division and, finally, with the type of tissue to which each of the lines used belongs. This content could be used for a future review on how biological issues affect the results, which are decisive in the design of the experiments.

Some of the division ranges for each of the parameters did not have data with which to carry out the statistics, so the statistics in these cases are incomplete.

No discrimination was made by exposure equipment, the type of coil used or laboratory conditions at the time of exposure (ambient magnetic fields), which could alter the results obtained based on the device used.

We consider that in order to draw conclusions about how parameters can determine the cellular response, it is not only important, as in this review, to make statistical groupings of effects but also to establish mechanisms of action that can explain how each of these parameters interacts with the cellular components.

## 5. Conclusions

This review arises from the need to consider each of the parameters of exposure to magnetic fields as an entity capable of producing a determined response in the cell. The importance that has been given to the intensity parameter since the establishment of regulations for the exposure of the general public to low-frequency electromagnetic fields has led to it also being the main parameter in in vitro experiments, considering intensity as the determining parameter for the occurrence of the main cellular effects and placing the “dose effect” as one of the main theories explaining the effects found. We must begin to think of combinations of exposure parameters that act at the cellular level as specific codes that give rise to specific responses. Biology, and more specifically cell biology, entails a set of rules that must be protected as a fundamental right. There must be a commitment on the part of scientific researchers in bioelectromagnetism to develop quality experiments based on rigorous working methodologies and the search for experimental conclusions that contribute to existing knowledge in this field in order to establish models of interaction between magnetic fields and biological systems that allow us to think of magnetic fields not only as a tool for modulating cell behaviour but also as a possible therapeutic application for various fields of medicine such as oncology, neurodegeneration or the healing of wounds and lesions.

## Figures and Tables

**Figure 1 ijms-25-05074-f001:**
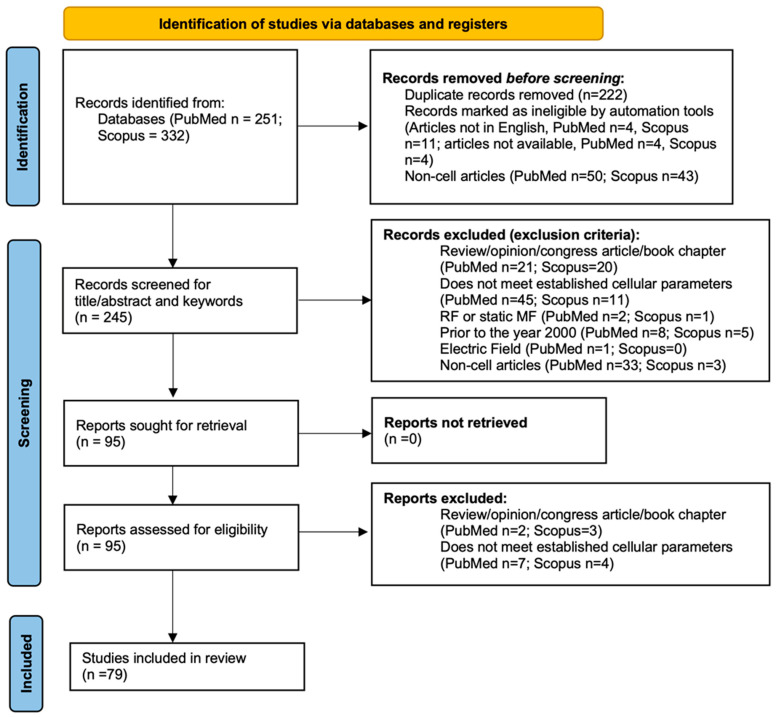
Identification diagram of databases and records following PRISMA-ScR protocol.

**Figure 2 ijms-25-05074-f002:**
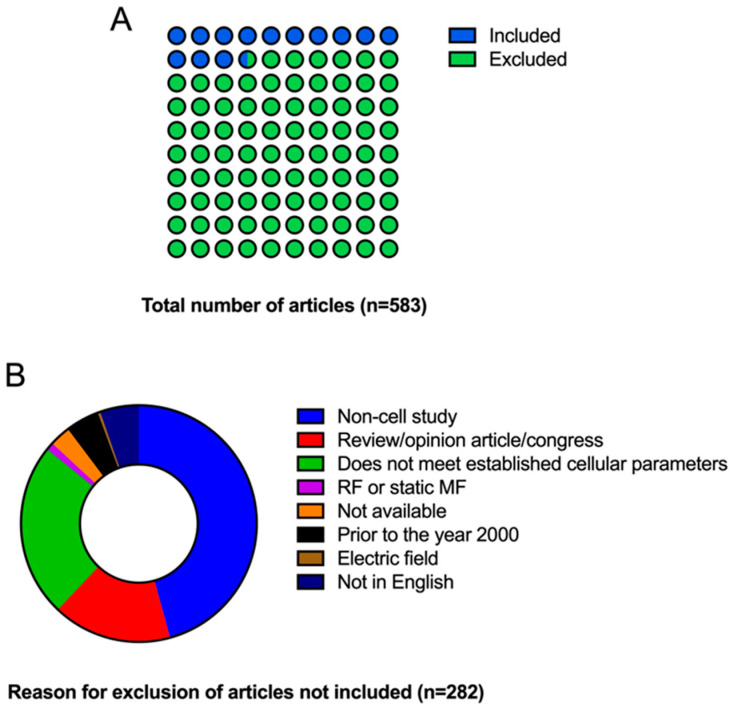
Diagrams representing items excluded and included in the review. (**A**) Total number of items in PubMed and Scopus according to search criteria. (**B**) Reason for exclusion of articles is not included (duplicate articles are not represented).

**Figure 3 ijms-25-05074-f003:**
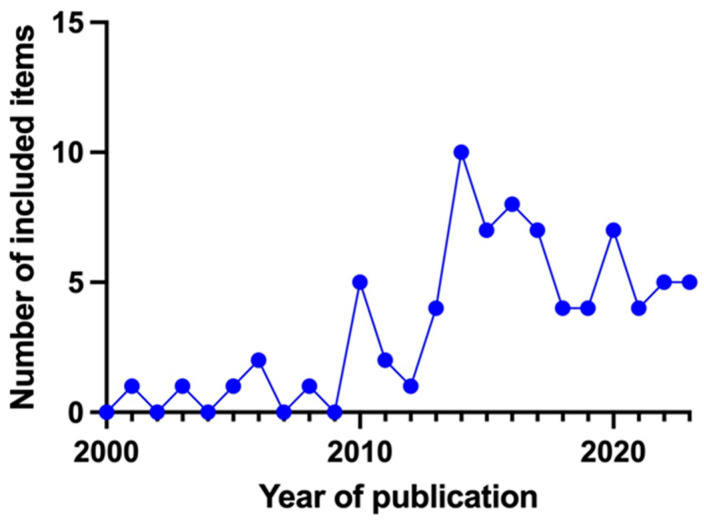
The temporal distribution of the articles included in the review by the year of publication.

**Figure 4 ijms-25-05074-f004:**
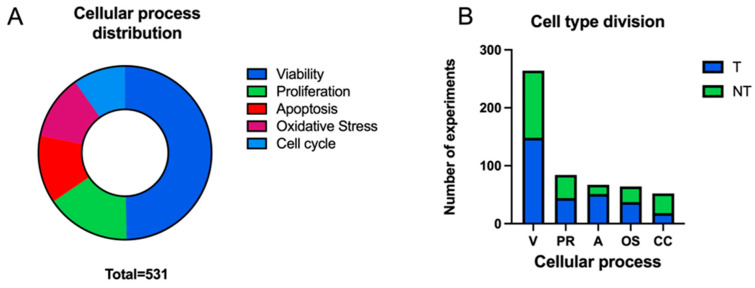
(**A**) The distribution of the cellular process studied over the total number of experiments analysed (n = 531). (**B**) The distribution of tumour (T)/non-tumour (NT) cells according to the cell process studied (V: viability, PR: proliferation, A: apoptosis, OS: oxidative stress/mitochondria, CC: cell cycle).

**Figure 5 ijms-25-05074-f005:**
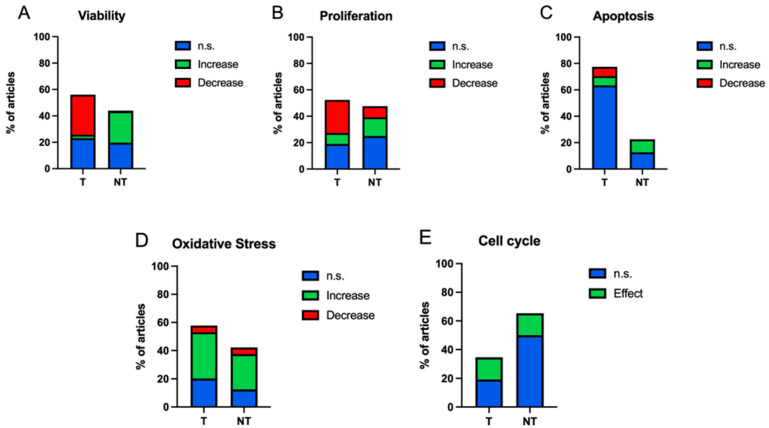
Bar charts of percentage of experiments with significant increase (green) and decrease (red), and non-significant (blue) results, differentiated by cell process ((**A**) viability, (**B**) proliferation, (**C**) apoptosis, (**D**) oxidative stress, (**E**) cell cycle) and cell type (T: tumour, NT: non-tumour).

**Figure 6 ijms-25-05074-f006:**
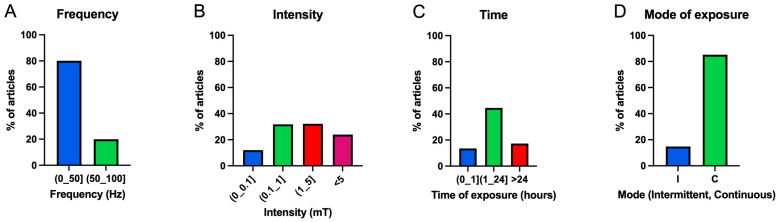
A bar chart of the percentage of experiments performed for each division of the different exposure parameters set: frequency (**A**), intensity (**B**), exposure time (**C**) and exposure mode (**D**) (I: intermittent; C: continuous).

**Figure 7 ijms-25-05074-f007:**
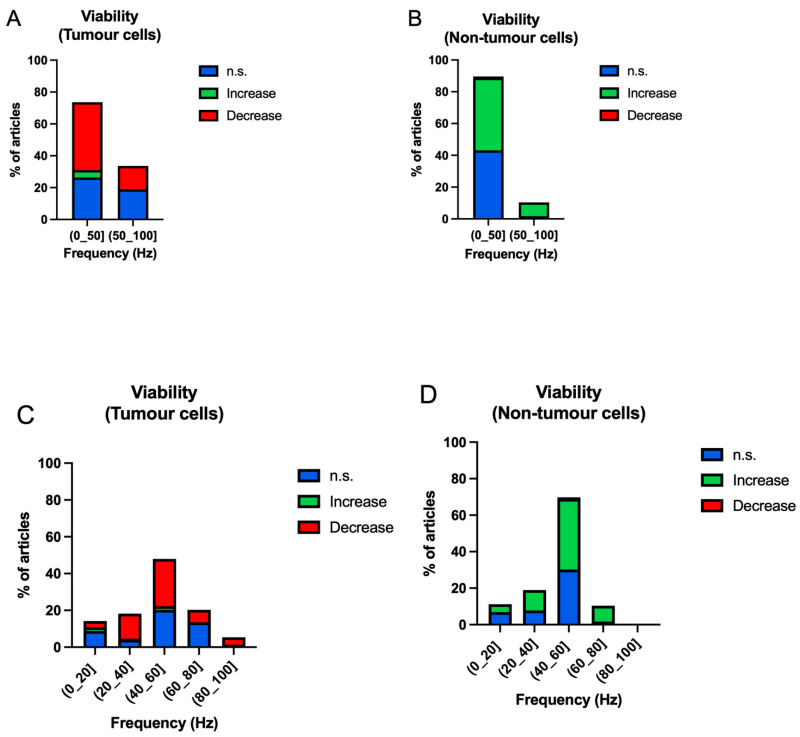
Bar charts of the occurrence of non-significant (blue), increase (green), decrease (red) results according to frequency bands in tumour and non-tumour cells in viability. (**A**,**B**) Frequencies below and above 50 Hz; (**C**,**D**) frequencies between 0 and 100 Hz in 20 Hz steps.

**Figure 8 ijms-25-05074-f008:**
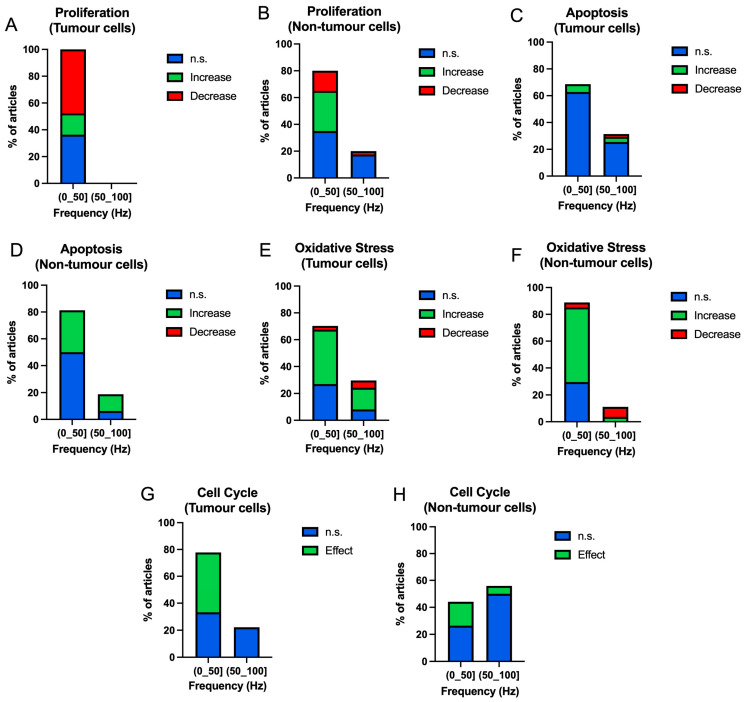
Bar graphs of the occurrence of non-significant (blue), increase (green), decrease (red) results according to frequency bands in tumour and non-tumour cells in different cellular processes ((**A**,**B**) proliferation; (**C**,**D**) apoptosis; (**E**,**F**) oxidative stress). Bar graphs of the occurrence of non-significant (blue) and significant (green) results according to frequency bands in tumour and non-tumour cells in cell cycle (**G**,**H**)).

**Figure 9 ijms-25-05074-f009:**
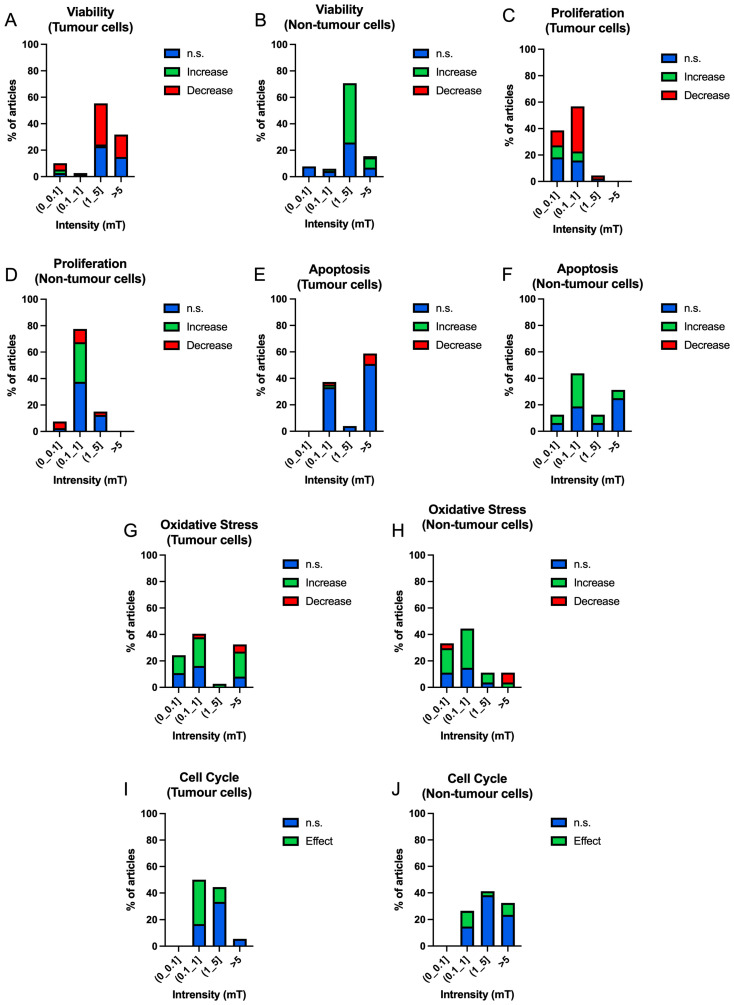
Bar charts of the occurrence of non-significant (blue), increase (green), decrease (red) results according to intensity bands in tumour and non-tumour cells in different cellular processes ((**A**,**B**) viability; (**C**,**D**) proliferation; (**E**,**F**) apoptosis; (**G**,**H**) oxidative stress. Bar graphs of the occurrence of non-significant (blue) and significant (green) results according to frequency bands in tumour and non-tumour cells in cell cycle (**I**,**J**)).

**Figure 10 ijms-25-05074-f010:**
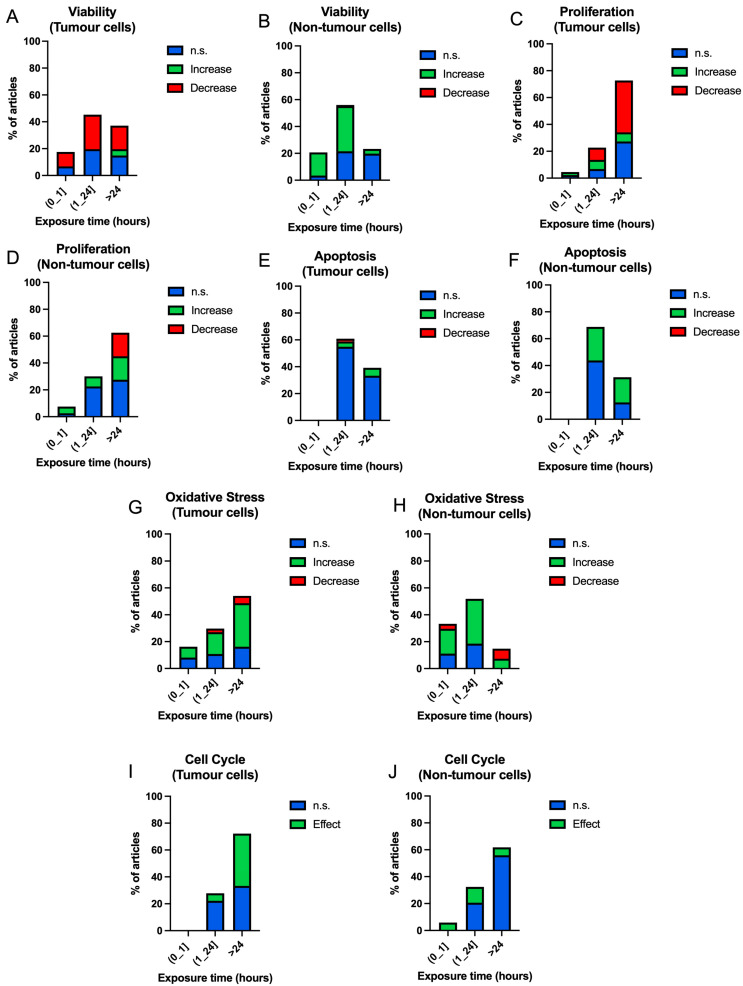
Bar charts of the occurrence of non-significant (blue), increase (green), decrease (red) results according to exposure time bands in tumour and non-tumour cells in different cellular processes ((**A**,**B**) viability; (**C**,**D**) proliferation; (**E**,**F**) apoptosis; (**G**,**H**) oxidative stress. Bar graphs of the occurrence of non-significant (blue) and significant (green) results according to frequency bands in tumour and non-tumour cells in cell cycle (**I**,**J**)).

**Figure 11 ijms-25-05074-f011:**
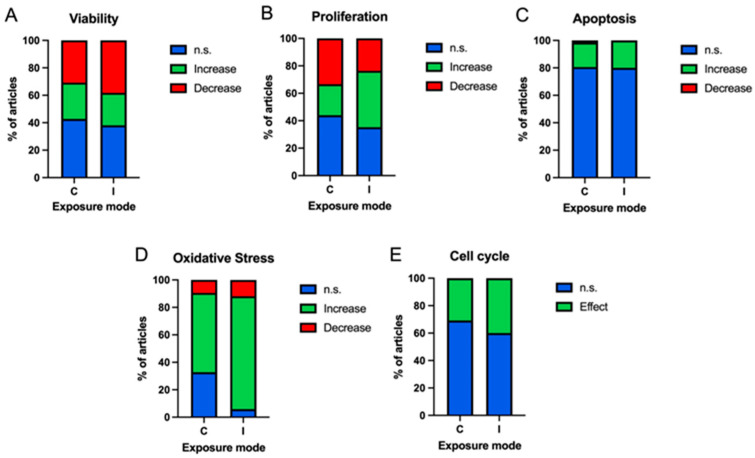
Bar charts of the occurrence of non-significant (blue), increase (green), decrease (red) results according to exposure mode (C: continuous/I: intermittent) in tumour and non-tumour cells in different cellular processes ((**A**) viability; (**B**) proliferation; (**C**) apoptosis; (**D**) oxidative stress. Bar graphs of the occurrence of non-significant (blue) and significant (green) results according to frequency bands in tumour and non-tumour cells in cell cycle (**E**)).

**Table 1 ijms-25-05074-t001:** A table containing the individual experiments of the articles selected according to the inclusion criteria for the analysis of the viability results in continuous exposure mode. Results: non-significant (0); decrease (1); increase (2). Increasing order by frequency.

ID	Authors	Frequency (Hz)	Intensity (mT)	Exposure Time (h)	Cell Type (T/NT)	Results	ID	Authors	Frequency (Hz)	Intensity (mT)	Exposure Time (h)	Cell Type (T/NT)	Results
1	Nezamtaheri et al. (2022) [74]	0.01	10	2	T	0	133	Koziorowska, A. et al. (2018) [75]	50	2	0.5	T	1
2	Nezamtaheri et al. (2022) [74]	0.01	10	2	NT	0	134	Koziorowska, A. et al. (2018) [75]	50	2	0.5	T	1
3	Nezamtaheri et al. (2022) [74]	0.01	10	2	T	0	135	Koziorowska, A. et al. (2018) [75]	50	2	0.5	NT	2
4	Nezamtaheri et al. (2022) [74]	0.01	10	2	T	0	136	Koziorowska, A. et al. (2018) [75]	50	2	0.5	NT	2
5	Koziorowska, A. et al. (2018) [75]	2	2	2	T	1	137	Koziorowska, A. et al. (2018) [75]	50	2	1	T	1
6	Ledda et al. (2018) [76]	7	0.0125	120	T	2	138	Koziorowska, A. et al. (2018) [75]	50	2	1	T	1
7	Ledda et al. (2018) [76]	7	0.0125	120	T	2	139	Koziorowska, A. et al. (2018) [75]	50	2	1	NT	2
8	García-Minguillán, O. et al. (2019) [77]	7.8	0.03	24	NT	0	140	Koziorowska, A. et al. (2018) [75]	50	2	1	NT	2
9	García-Minguillán, O. et al. (2019) [77]	7.8	0.03	24	NT	0	141	Koziorowska, A. et al. (2018) [75]	50	2	2	T	1
10	García-Minguillán, O. et al. (2019) [77]	14	0.03	24	T	0	142	Koziorowska, A. et al. (2018) [75]	50	2	2	T	1
11	García-Minguillán, O. et al. (2020) [78]	20	0.03	24	T	1	143	Koziorowska, A. et al. (2018) [75]	50	2	2	NT	2
12	García-Minguillán, O. et al. (2020) [78]	20	0.1	24	T	0	144	Koziorowska, A. et al. (2018) [75]	50	2	2	NT	0
13	García-Minguillán, O. et al. (2020) [78]	20	0.1	48	T	2	145	Koziorowska, A. et al. (2018) [75]	50	2	3	T	1
14	García-Minguillán, O. et al. (2020) [78]	20	0.1	72	T	0	146	Koziorowska, A. et al. (2018) [75]	50	2	3	T	1
15	Koziorowska, A. et al. (2018) [75]	20	2	2	T	1	147	Koziorowska, A. et al. (2018) [75]	50	2	3	NT	2
16	Koziorowska, A. et al. (2018) [75]	20	2	2	T	0	148	Koziorowska, A. et al. (2018) [75]	50	2	3	NT	0
17	Koziorowska, A. et al. (2018) [75]	20	2	2	NT	2	149	Koziorowska, A. et al. (2018) [75]	50	2.5	0.5	T	0
18	Koziorowska, A. et al. (2018) [75]	20	2	2	NT	0	150	Koziorowska, A. et al. (2018) [75]	50	2.5	0.5	T	1
19	Koziorowska, A. et al. (2018) [75]	20	2.5	2	T	1	151	Koziorowska, A. et al. (2018) [75]	50	2.5	0.5	NT	2
20	Koziorowska, A. et al. (2018) [75]	20	2.5	2	T	0	152	Koziorowska, A. et al. (2018) [75]	50	2.5	0.5	NT	2
21	Koziorowska, A. et al. (2018) [75]	20	2.5	2	NT	2	153	Koziorowska, A. et al. (2018) [75]	50	2.5	1	T	1
22	Koziorowska, A. et al. (2018) [75]	20	2.5	2	NT	0	154	Koziorowska, A. et al. (2018) [75]	50	2.5	1	T	1
23	Koziorowska, A. et al. (2018) [75]	20	3	2	T	0	155	Koziorowska, A. et al. (2018) [75]	50	2.5	1	NT	2
24	Koziorowska, A. et al. (2018) [75]	20	3	2	T	0	156	Koziorowska, A. et al. (2018) [75]	50	2.5	1	NT	2
25	Koziorowska, A. et al. (2018) [75]	20	3	2	NT	2	157	Koziorowska, A. et al. (2018) [75]	50	2.5	2	T	1
26	Koziorowska, A. et al. (2018) [75]	20	3	2	NT	0	158	Koziorowska, A. et al. (2018) [75]	50	2.5	2	T	1
27	Koziorowska, A. et al. (2018) [75]	20	4	2	T	0	159	Koziorowska, A. et al. (2018) [75]	50	2.5	2	NT	2
28	Koziorowska, A. et al. (2018) [75]	20	4	2	T	0	160	Koziorowska, A. et al. (2018) [75]	50	2.5	2	NT	0
29	Koziorowska, A. et al. (2018) [75]	20	4	2	NT	2	161	Koziorowska, A. et al. (2018) [75]	50	2.5	3	T	1
30	Koziorowska, A. et al. (2018) [75]	20	4	2	NT	0	162	Koziorowska, A. et al. (2018) [75]	50	2.5	3	T	1
31	Koziorowska, A. et al. (2018) [75]	20	6	2	T	0	163	Koziorowska, A. et al. (2018) [75]	50	2.5	3	NT	2
32	Koziorowska, A. et al. (2018) [75]	20	6	2	T	1	164	Koziorowska, A. et al. (2018) [75]	50	2.5	3	NT	2
33	Koziorowska, A. et al. (2018) [75]	20	6	2	NT	2	165	Koziorowska, A. et al. (2018) [75]	50	3	0.5	T	0
34	Koziorowska, A. et al. (2018) [75]	20	6	2	NT	0	166	Koziorowska, A. et al. (2018) [75]	50	3	0.5	T	1
35	Wang, M. et al. (2021) [79]	23.49	0.5	48	T	1	167	Koziorowska, A. et al. (2018) [75]	50	3	0.5	NT	2
36	Wang, M. et al. (2021) [79]	23.49	0.5	48	T	1	168	Koziorowska, A. et al. (2018) [75]	50	3	0.5	NT	2
37	Wang, M. et al. (2021) [79]	23.49	0.5	48	NT	2	169	Koziorowska, A. et al. (2018) [75]	50	3	1	T	1
38	García-Minguillán, O. et al. (2020) [78]	30	0.1	24	T	1	170	Koziorowska, A. et al. (2018) [75]	50	3	1	T	1
39	García-Minguillán, O. et al. (2020) [78]	30	0.1	48	T	1	171	Koziorowska, A. et al. (2018) [75]	50	3	1	NT	2
40	García-Minguillán, O. et al. (2020) [78]	30	0.1	72	T	1	172	Koziorowska, A. et al. (2018) [75]	50	3	1	NT	2
41	Koziorowska, A. et al. (2018) [75]	30	2	2	T	1	173	Koziorowska, A. et al. (2018) [75]	50	3	2	T	1
42	Koziorowska, A. et al. (2018) [75]	30	2	2	T	1	174	Koziorowska, A. et al. (2018) [75]	50	3	2	T	1
43	Koziorowska, A. et al. (2018) [75]	30	2	2	NT	0	175	Koziorowska, A. et al. (2018) [75]	50	3	2	NT	2
44	Koziorowska, A. et al. (2018) [75]	30	2	2	NT	2	176	Koziorowska, A. et al. (2018) [75]	50	3	2	NT	2
45	Koziorowska, A. et al. (2018) [75]	30	2.5	2	T	1	177	Koziorowska, A. et al. (2018) [75]	50	3	3	T	1
46	Koziorowska, A. et al. (2018) [75]	30	2.5	2	T	1	178	Koziorowska, A. et al. (2018) [75]	50	3	3	T	1
47	Koziorowska, A. et al. (2018) [75]	30	2.5	2	NT	2	179	Koziorowska, A. et al. (2018) [75]	50	3	3	NT	2
48	Koziorowska, A. et al. (2018) [75]	30	2.5	2	NT	2	180	Koziorowska, A. et al. (2018) [75]	50	3	3	NT	0
49	Koziorowska, A. et al. (2018) [75]	30	3	2	T	1	181	Koziorowska, A. et al. (2018) [75]	50	4	0.5	T	0
50	Koziorowska, A. et al. (2018) [75]	30	3	2	T	1	182	Koziorowska, A. et al. (2018) [75]	50	4	0.5	T	1
51	Koziorowska, A. et al. (2018) [75]	30	3	2	NT	2	183	Koziorowska, A. et al. (2018) [75]	50	4	0.5	NT	2
52	Koziorowska, A. et al. (2018) [75]	30	3	2	NT	2	184	Koziorowska, A. et al. (2018) [75]	50	4	0.5	NT	2
53	Koziorowska, A. et al. (2018) [75]	30	4	2	T	1	185	Koziorowska, A. et al. (2018) [75]	50	4	1	T	1
54	Koziorowska, A. et al. (2018) [75]	30	4	2	T	1	186	Koziorowska, A. et al. (2018) [75]	50	4	1	T	1
55	Koziorowska, A. et al. (2018) [75]	30	4	2	NT	0	187	Koziorowska, A. et al. (2018) [75]	50	4	1	NT	2
56	Koziorowska, A. et al. (2018) [75]	30	4	2	NT	2	188	Koziorowska, A. et al. (2018) [75]	50	4	1	NT	2
57	Koziorowska, A. et al. (2018) [75]	30	6	2	T	0	189	Koziorowska, A. et al. (2018) [75]	50	4	2	T	1
58	Koziorowska, A. et al. (2018) [75]	30	6	2	T	1	190	Koziorowska, A. et al. (2018) [75]	50	4	2	T	1
59	Koziorowska, A. et al. (2018) [75]	30	6	2	NT	0	191	Koziorowska, A. et al. (2018) [75]	50	4	2	NT	2
60	Koziorowska, A. et al. (2018) [75]	30	6	2	NT	0	192	Koziorowska, A. et al. (2018) [75]	50	4	2	NT	0
61	Wang, M. et al. (2021) [79]	39.15	0.5	48	T	0	193	Koziorowska, A. et al. (2018) [75]	50	4	3	T	1
62	Wang, M. et al. (2021) [79]	39.15	0.5	48	T	2	194	Koziorowska, A. et al. (2018) [75]	50	4	3	T	1
63	Wang, M. et al. (2021) [79]	39.15	0.5	48	NT	2	195	Koziorowska, A. et al. (2018) [75]	50	4	3	NT	2
64	Koziorowska, A. et al. (2018) [75]	40	2	2	T	1	196	Koziorowska, A. et al. (2018) [75]	50	4	3	NT	2
65	Koziorowska, A. et al. (2018) [75]	40	2	2	T	1	197	Koziorowska, A. et al. (2018) [75]	50	6	0.5	T	0
66	Koziorowska, A. et al. (2018) [75]	40	2	2	NT	2	198	Koziorowska, A. et al. (2018) [75]	50	6	0.5	T	1
67	Koziorowska, A. et al. (2018) [75]	40	2	2	NT	0	199	Koziorowska, A. et al. (2018) [75]	50	6	0.5	NT	2
68	Koziorowska, A. et al. (2018) [75]	40	2.5	2	T	0	200	Koziorowska, A. et al. (2018) [75]	50	6	0.5	NT	2
69	Koziorowska, A. et al. (2018) [75]	40	2.5	2	T	1	201	Koziorowska, A. et al. (2018) [75]	50	6	1	T	1
70	Koziorowska, A. et al. (2018) [75]	40	2.5	2	NT	2	202	Koziorowska, A. et al. (2018) [75]	50	6	1	T	1
71	Koziorowska, A. et al. (2018) [75]	40	2.5	2	NT	0	203	Koziorowska, A. et al. (2018) [75]	50	6	1	NT	2
72	Koziorowska, A. et al. (2018) [75]	40	3	2	T	0	204	Koziorowska, A. et al. (2018) [75]	50	6	1	NT	2
73	Koziorowska, A. et al. (2018) [75]	40	3	2	T	1	205	Koziorowska, A. et al. (2018) [75]	50	6	2	T	1
74	Koziorowska, A. et al. (2018) [75]	40	3	2	NT	2	206	Koziorowska, A. et al. (2018) [75]	50	6	2	T	1
75	Koziorowska, A. et al. (2018) [75]	40	3	2	NT	0	207	Koziorowska, A. et al. (2018) [75]	50	6	2	NT	2
76	Koziorowska, A. et al. (2018) [75]	40	4	2	T	0	208	Koziorowska, A. et al. (2018) [75]	50	6	2	NT	0
77	Koziorowska, A. et al. (2018) [75]	40	4	2	T	1	209	Koziorowska, A. et al. (2018) [75]	50	6	3	T	0
78	Koziorowska, A. et al. (2018) [75]	40	4	2	NT	2	210	Koziorowska, A. et al. (2018) [75]	50	6	3	T	1
79	Koziorowska, A. et al. (2018) [75]	40	4	2	NT	0	211	Koziorowska, A. et al. (2018) [75]	50	6	3	NT	0
80	Koziorowska, A. et al. (2018) [75]	40	6	2	T	0	212	Koziorowska, A. et al. (2018) [75]	50	6	3	NT	0
81	Koziorowska, A. et al. (2018) [75]	40	6	2	T	1	213	Yin, C. et al. (2016) [80]	50	8	1.5	NT	1
82	Koziorowska, A. et al. (2018) [75]	40	6	2	NT	2	214	García-Minguillán, O. et al. (2019) [77]	51	0.03	24	T	1
83	Koziorowska, A. et al. (2018) [75]	40	6	2	NT	0	215	Cho, S. et al. (2014) [81]	60	0.8	24	NT	0
84	García-Minguillán, O. et al. (2019) [77]	45	0.03	24	T	1	216	Cho, S. et al. (2014) [81]	60	0.8	48	NT	0
85	De Groot, M. et al. (2016) [82]	50	0.001	168	NT	0	217	Koziorowska, A. et al. (2018) [75]	60	2	2	T	0
86	De Groot, M. et al. (2016) [82]	50	0.01	168	NT	0	218	Koziorowska, A. et al. (2018) [75]	60	2	2	T	0
87	Calcabrini, C. et al. (2017) [83]	50	0.025	1	NT	0	219	Koziorowska, A. et al. (2018) [75]	60	2	2	NT	2
88	Calcabrini, C. et al. (2017) [83]	50	0.05	1	NT	0	220	Koziorowska, A. et al. (2018) [75]	60	2	2	NT	2
89	Calcabrini, C. et al. (2017) [83]	50	0.1	1	NT	0	221	Koziorowska, A. et al. (2018) [75]	60	2.5	2	T	0
90	De Groot, M. et al. (2016) [82]	50	0.1	168	NT	0	222	Koziorowska, A. et al. (2018) [75]	60	2.5	2	T	0
91	García-Minguillán, O. et al. (2020) [78]	50	0.1	24	T	1	223	Koziorowska, A. et al. (2018) [75]	60	2.5	2	NT	2
92	García-Minguillán, O. et al. (2020) [78]	50	0.1	48	T	0	224	Koziorowska, A. et al. (2018) [75]	60	2.5	2	NT	2
93	García-Minguillán, O. et al. (2020) [78]	50	0.1	72	T	2	225	Koziorowska, A. et al. (2018) [75]	60	3	2	T	0
94	Morabito, C. et al. (2017) [84]	50	0.1	168	NT	0	226	Koziorowska, A. et al. (2018) [75]	60	3	2	T	0
95	Calcabrini, C. et al. (2017) [83]	50	0.2	1	NT	0	227	Koziorowska, A. et al. (2018) [75]	60	3	2	NT	2
96	Morabito, C. et al. (2017) [84]	50	0.5	168	NT	0	228	Koziorowska, A. et al. (2018) [75]	60	3	2	NT	2
97	Ma, Q. et al. (2014) [85]	50	0.5	72	NT	0	229	Koziorowska, A. et al. (2018) [75]	60	4	2	T	0
98	Ma, Q. et al. (2014) [85]	50	1	72	NT	0	230	Koziorowska, A. et al. (2018) [75]	60	4	2	T	0
99	Costantini, E. et al. (2022) [86]	50	1	6	NT	2	231	Koziorowska, A. et al. (2018) [75]	60	4	2	NT	2
100	Costantini, E. et al. (2022) [86]	50	1	24	NT	2	232	Koziorowska, A. et al. (2018) [75]	60	4	2	NT	2
101	Costantini, E. et al. (2022) [86]	50	1	48	NT	2	233	Koziorowska, A. et al. (2018) [75]	60	6	2	T	0
102	Morabito, C. et al. (2017) [84]	50	1	168	NT	0	234	Koziorowska, A. et al. (2018) [75]	60	6	2	T	0
103	De Groot, M. et al. (2016) [82]	50	1	168	NT	0	235	Koziorowska, A. et al. (2018) [75]	60	6	2	NT	2
104	Duan, W. et al. (2015) [87]	50	1	24	NT	0	236	Koziorowska, A. et al. (2018) [75]	60	6	2	NT	2
105	Park, J. et al. (2013) [88]	50	1	96	NT	0	237	Song, K. et al. (2018) [89]	60	6	0.5	T	0
106	Park, J. et al. (2013) [88]	50	1	192	NT	0	238	Song, K. et al. (2018) [89]	60	6	0.5	T	0
107	Garip, A. et al. (2010) [90]	50	1	3	T	0	239	Song, K. et al. (2018) [89]	60	6	2.5	T	0
108	Vianale, G. et al. (2008) [91]	50	1	4	NT	0	240	Song, K. et al. (2018) [89]	60	6	2.5	T	0
109	Vianale, G. et al. (2008) [91]	50	1	12	NT	0	241	Song, K. et al. (2018) [89]	60	6	24	T	0
110	Vianale, G. et al. (2008) [91]	50	1	24	NT	0	242	Song, K. et al. (2018) [89]	60	6	24	T	0
111	Vianale, G. et al. (2008) [91]	50	1	48	NT	0	243	Song, K. et al. (2018) [89]	60	6	48	T	1
112	Vianale, G. et al. (2008) [91]	50	1	72	NT	0	244	Song, K. et al. (2018) [89]	60	6	48	T	1
113	Vianale, G. et al. (2008) [91]	50	1	96	NT	0	245	Song, K. et al. (2018) [89]	60	6	72	T	1
114	Morabito, C. et al. (2010) [92]	50	1	0.5	T	0	246	Song, K. et al. (2018) [89]	60	6	72	T	1
115	Morabito, C. et al. (2010) [92]	50	1	24	T	0	247	Song, K. et al. (2018) [89]	60	6	96	T	0
116	Morabito, C. et al. (2010) [92]	50	1	48	T	0	248	Song, K. et al. (2018) [89]	60	6	96	T	0
117	Morabito, C. et al. (2010) [92]	50	1	72	T	0	249	Song, K. et al. (2018) [89]	60	6	96	T	0
118	Morabito, C. et al. (2010) [92]	50	1	96	T	0	250	Song, K. et al. (2018) [89]	60	6	120	T	0
119	Morabito, C. et al. (2010) [92]	50	1	144	T	0	251	Song, K. et al. (2018) [89]	60	6	120	T	1
120	Morabito, C. et al. (2010) [92]	50	1	192	T	0	252	Song, K. et al. (2018) [89]	60	6	144	T	1
121	Reale, M. et al. (2014) [93]	50	1	1	T	0	253	Song, K. et al. (2018) [89]	60	6	144	T	1
122	Reale, M. et al. (2014) [93]	50	1	3	T	0	254	Song, K. et al. (2018) [89]	60	6	144	T	1
123	Reale, M. et al. (2014) [93]	50	1	6	T	0	255	Song, K. et al. (2018) [89]	60	6	168	T	1
124	Reale, M. et al. (2014) [93]	50	1	24	T	0	256	Song, K. et al. (2018) [89]	60	6	168	T	1
125	Falone, S. et al. (2007) [94]	50	1	24	T	0	257	Akbarnejad, Z. et al. (2017) [95]	100	10	72	T	0
126	Falone, S. et al. (2007) [94]	50	1	48	T	0	258	Akbarnejad, Z. et al. (2017) [95]	100	10	72	T	0
127	Falone, S. et al. (2007) [94]	50	1	72	T	2	259	Akbarnejad, Z. et al. (2017) [95]	100	10	96	T	1
128	Falone, S. et al. (2007) [94]	50	1	96	T	2	260	Akbarnejad, Z. et al. (2017) [95]	100	10	96	T	1
129	Ma, Q. et al. (2014) [85]	50	2	24	NT	0	261	Akbarnejad, Z. et al. (2017) [95]	100	10	120	T	1
130	Ma, Q. et al. (2014) [85]	50	2	48	NT	0	262	Akbarnejad, Z. et al. (2017) [95]	100	10	120	T	1
131	Ma, Q. et al. (2014) [85]	50	2	72	NT	0	263	Akbarnejad, Z. et al. (2017) [95]	100	10	144	T	1
132	Barati, A. et al. (2023) [96]	50	2	72	T	1	264	Akbarnejad, Z. et al. (2017) [95]	100	10	144	T	1

**Table 2 ijms-25-05074-t002:** A table containing the individual experiments of the articles selected according to the inclusion criteria for the analysis of the intermittent exposure mode. Results: non-significant (0); decrease (1); increase (2).

ID	Authors	Cell Type (T/NT)	Results	ID	Authors	Cell Type (T/NT)	Results
VIABILITY				27	Rahimi, S. et al. (2023) [97]	NT	0
1	Wang, M. et al. (2021) [79]	T	0	28	Razavi, S. et al. (2013) [98]	NT	2
2	Wang, M. et al. (2021) [79]	T	1	29	Razavi, S. et al. (2013) [98]	NT	2
3	Wang, M. et al. (2021) [79]	NT	0	30	Razavi, S. et al. (2013) [98]	NT	2
4	Wang, M. et al. (2021) [79]	T	0	31	Razavi, S. et al. (2013) [98]	NT	0
5	Wang, M. et al. (2021) [79]	T	1	32	Focke, F. et al. (2010) [99]	NT	1
6	Wang, M. et al. (2021) [79]	NT	0	33	Focke, F. et al. (2010) [99]	T	0
7	Wang, M. et al. (2021) [79]	T	0	34	Grant, D. et al. (2014) [100]	NT	0
8	Wang, M. et al. (2021) [79]	T	1	APOPTOSIS			
9	Wang, M. et al. (2021) [79]	NT	2	1	Cios, A. et al. (2021) [101]	NT	0
10	Ross, C. et al. (2018) [102]	NT	0	2	Cios, A. et al. (2021) [101]	T	2
11	García-Minguillán, O. et al. (2019) [77]	T	0	3	Cios, A. et al. (2021) [101]	NT	0
12	García-Minguillán, O. et al. (2019) [77]	T	1	4	Cios, A. et al. (2021) [101]	T	0
13	Barati, A. et al. (2021) [96]	T	1	5	Liu, Y. et al. (2015) [103]	NT	0
14	Samiei, M. et al. (2020) [104]	NT	2	6	Liu, Y. et al. (2015) [103]	NT	0
15	Samiei, M. et al. (2020) [104]	NT	2	7	Liu, Y. et al. (2015) [103]	NT	0
16	Samiei, M. et al. (2020) [104]	NT	2	8	Ma, Q. et al. (2016) [105]	NT	0
17	Samiei, M. et al. (2020) [104]	NT	0	9	Focke, F. et al. (2010) [99]	NT	2
18	Cios, A. et al. (2021) [101]	NT	2	10	Focke, F. et al. (2010) [99]	T	0
19	Cios, A. et al. (2021) [101]	T	1	OXIDATIVE STRESS/MITOCHONDRIA			
20	Cios, A. et al. (2021) [101]	NT	1	1	Cios, A. et al. (2021) [101]	NT	2
21	Cios, A. et al. (2021) [101]	T	1	2	Cios, A. et al. (2021) [101]	T	2
PROLIFERATION				3	Cios, A. et al. (2021) [101]	NT	2
1	Mehdizadeh, R. et al. (2023) [106]		1	4	Cios, A. et al. (2021) [101]	T	2
2	Mehdizadeh, R. et al. (2023) [106]	T	0	5	Lekovic, M. et al. (2020) [107]	NT	1
3	Mehdizadeh, R. et al. (2023) [106]	T	1	6	Lekovic, M. et al. (2020) [107]	NT	0
4	Ross, C. et al. (2018) [102]	T	0	7	Lekovic, M. et al. (2020) [107]	NT	1
5	Ross, C. et al. (2018) [102]	NT	0	8	Lekovic, M. et al. (2020) [107]	NT	2
6	Fathi, E. et al. (2017) [108]	NT	1	9	Ayse, I. et al. (2010) [109]	T	2
7	Lekovic, M. et al. (2020) [107]	NT	0	10	Choi, J. et al. (2022) [110]	NT	2
8	Lekovic, M. et al. (2020) [107]	NT	2	11	Choi, J. et al. (2022) [110]	NT	2
9	Lekovic, M. et al. (2020) [107]	NT	0	12	Choi, J. et al. (2022) [110]	NT	2
10	Lekovic, M. et al. (2020) [107]	NT	1	13	Choi, J. et al. (2022) [110]	NT	2
11	Ma, Q. et al. (2016) [105]	NT	0	14	Choi, J. et al. (2022) [110]	NT	2
12	Ma, Q. et al. (2016) [105]	NT	0	15	Choi, J. et al. (2022) [110]	NT	2
13	Ma, Q. et al. (2016) [105]	NT	2	16	Ki, G. et al. (2020) [111]	NT	2
14	Restrepo, A.F. et al. (2016) [112]	NT	0	17	Falone, S. et al. (2016) [113]	T	2
15	Restrepo, A.F. et al. (2016) [112]	NT	2	CELL CYCLE			
16	Restrepo, A.F. et al. (2016) [112]	NT	2	1	Cios, A. et al. (2021) [101]	NT	0
17	Restrepo, A.F. et al. (2016) [112]	NT	2	2	Cios, A. et al. (2021) [101]	T	1
18	Restrepo, A.F. et al. (2016) [112]	NT	2	3	Cios, A. et al. (2021) [101]	NT	1
19	Restrepo, A.F. et al. (2016) [112]	NT	2	4	Cios, A. et al. (2021) [101]	T	1
20	Restrepo, A.F. et al. (2016) [112]	NT	2	5	Fan, W. et al. (2015) [114]	NT	1
21	Restrepo, A.F. et al. (2016) [112]	NT	2	6	Liu, Y. et al. (2015) [103]	NT	0
22	Restrepo, A.F. et al. (2016) [112]	NT	2	7	Liu, Y. et al. (2015) [103]	NT	0
23	Restrepo, A.F. et al. (2016) [112]	NT	2	8	Liu, Y. et al. (2015) [103]	NT	0
24	Rahimi, S. et al. (2023) [97]	NT	2	9	Focke, F. et al. (2010) [99]	NT	0
25	Rahimi, S. et al. (2023) [97]	NT	1	10	Focke, F. et al. (2010) [99]	T	0
26	Rahimi, S. et al. (2023) [97]	NT	1				

**Table 3 ijms-25-05074-t003:** A table containing the individual experiments of the articles selected according to the inclusion criteria for the analysis of the proliferation results in continuous exposure mode. Results: non-significant (0); decrease (1); increase (2). Increasing order by frequency.

ID	Authors	Frequency (Hz)	Intensity (mT)	Exposure Time (h)	Cell Type (T/NT)	Results	ID	Authors	Frequency (Hz)	Intensity (mT)	Exposure Time (h)	Cell Type (T/NT)	Results
1	Nezamtaheri, M. et al. (2022) [74]	0.01	1	2	T	1	43	Cheng, Y. et al. (2015) [115]	50	0.4	4	NT	0
2	Nezamtaheri, M. et al. (2022) [74]	0.01	1	120	T	1	44	Cheng, Y. et al. (2015) [115]	50	0.4	8	NT	0
3	Nezamtaheri, M. et al. (2022) [74]	0.01	1	120	NT	1	45	Cheng, Y. et al. (2015) [115]	50	0.4	16	NT	2
4	Nezamtaheri, M. et al. (2022) [74]	0.01	1	120	T	1	46	Cheng, Y. et al. (2015) [115]	50	0.4	24	NT	2
5	Nezamtaheri, M. et al. (2022) [74]	1	0.1	120	T	1	47	Cheng, Y. et al. (2015) [115]	50	0.4	32	NT	2
6	Bergandi, L. et al. (2022) [116]	3	0.115	48	T	1	48	Rezaie-Tavirani, M. et al. (2017) [117]	50	0.5	3	T	1
7	Bergandi, L. et al. (2022) [116]	4	0.115	48	T	1	49	Morabito, C. et al. (2017) [84]	50	0.5	168	NT	0
8	Bergandi, L. et al. (2022) [116]	6	0.115	48	T	1	50	Patruno, A. et al. (2015) [118]	50	1	1	NT	2
9	Bergandi, L. et al. (2022) [116]	6	0.115	48	T	0	51	Patruno, A. et al. (2015) [118]	50	1	24	NT	2
10	Bergandi, L. et al. (2022) [116]	6	0.115	96	T	1	52	Rezaie-Tavirani, M. et al. (2017) [117]	50	1	3	T	1
11	Bergandi, L. et al. (2022) [116]	6	0.115	96	T	0	53	Oh, I. et al. (2020) [119]	50	1	120	T	1
12	Ledda, M. et al. (2018) [76]	7	0.0125	120	NT	1	54	Morabito, C. et al. (2017) [84]	50	1	168	NT	0
13	Ledda, M. et al. (2018) [76]	7	0.0125	120	NT	1	55	Vianale, G. et al. (2008) [91]	50	1	4	NT	0
14	Wang, M. et al. (2021) [79]	7.83	0.5	48	T	1	56	Vianale, G. et al. (2008) [91]	50	1	12	NT	0
15	Wang, M. et al. (2021) [79]	7.83	0.5	48	T	1	57	Vianale, G. et al. (2008) [91]	50	1	24	NT	0
16	Wang, M. et al. (2021) [79]	7.83	0.5	48	NT	1	58	Vianale, G. et al. (2008) [91]	50	1	48	NT	2
17	Bergandi, L. et al. (2022) [116]	8	115	96	T	0	59	Vianale, G. et al. (2008) [91]	50	1	72	NT	2
18	Bergandi, L. et al. (2022) [116]	8	115	96	T	1	60	Vianale, G. et al. (2008) [91]	50	1	96	NT	2
19	Bergandi, L. et al. (2022) [116]	10	115	48	T	1	61	Wolf, F. et al. (2005) [120]	50	1	48	T	2
20	Bergandi, L. et al. (2022) [116]	14	115	48	T	1	62	Wolf, F. et al. (2005) [120]	50	1	49	NT	2
21	Ruiz Gómez, M.J. et al. (2000) [121]	25	1.5	2.75	T	0	63	Wolf, F. et al. (2005) [120]	50	1	50	NT	2
22	Ruiz Gómez, M.J. et al. (2000) [121]	25	1.5	2.75	T	1	64	Kim, H. et al. (2013) [122]	50	1	48	NT	0
23	Destefanis, M. et al. (2015) [123]	50	0.045	168	T	1	65	Kim, H. et al. (2013) [122]	50	1	144	NT	1
24	Destefanis, M. et al. (2015) [123]	50	0.045	168	T	1	66	Kim, H. et al. (2013) [122]	50	1	288	NT	1
25	Destefanis, M. et al. (2015) [123]	50	0.045	168	T	1	67	Morabito, C. et al. (2017) [84]	50	1	0.5	NT	0
26	Destefanis, M. et al. (2015) [123]	50	0.045	168	T	1	68	Morabito, C. et al. (2017) [84]	50	1	24	NT	0
27	Morabito, C. et al. (2017) [84]	50	0.1	168	NT	0	69	Morabito, C. et al. (2017) [84]	50	1	48	NT	0
28	Morabito, C. et al. (2010) [92]	50	0.1	0.5	T	0	70	Morabito, C. et al. (2017) [84]	50	1	72	NT	2
29	Morabito, C. et al. (2010) [92]	50	0.1	24	T	0	71	Morabito, C. et al. (2017) [84]	50	1	96	NT	0
30	Morabito, C. et al. (2010) [92]	50	0.1	48	T	0	72	Morabito, C. et al. (2017) [84]	50	1	120	NT	0
31	Morabito, C. et al. (2010) [92]	50	0.1	72	T	2	73	Falone, S. et al. (2007) [94]	50	1	24	T	0
32	Morabito, C. et al. (2010) [92]	50	0.1	96	T	0	74	Falone, S. et al. (2007) [94]	50	1	48	T	0
33	Morabito, C. et al. (2010) [92]	50	0.1	120	T	0	75	Falone, S. et al. (2007) [94]	50	1	72	T	0
34	Morabito, C. et al. (2010) [92]	50	0.1	144	T	0	76	Falone, S. et al. (2007) [94]	50	1	96	T	0
35	Morabito, C. et al. (2010) [92]	50	0.1	168	T	0	77	Kim, S. et al. (2017) [124]	60	0.8	10	NT	0
36	Morabito, C. et al. (2010) [92]	50	0.1	192	T	0	78	Kim, S. et al. (2017) [124]	60	0.8	20	NT	0
37	Srdjenovic, B. et al. (2014) [125]	50	0.1	3	T	2	79	Huang, C. et al. (2014) [126]	60	1.5	24	NT	0
38	Srdjenovic, B. et al. (2014) [125]	50	0.1	24	T	2	80	Huang, C. et al. (2014) [126]	60	1.5	48	NT	0
39	Srdjenovic, B. et al. (2014) [125]	50	0.1	48	T	2	81	Huang, C. et al. (2014) [126]	60	1.5	72	NT	0
40	Qiu, L. et al. (2018) [127]	50	0.4	1	NT	2	82	Huang, C. et al. (2014) [126]	60	1.5	96	NT	0
41	Chen, L. et al. (2020) [128]	50	0.4	1	T	2	83	Huang, C. et al. (2014) [126]	60	1.5	120	NT	0
42	Nezamtaheri, M. et al. (2022) [74]	0.01	1	2	T	1	84	Huang, C. et al. (2014) [126]	60	1.5	144	NT	1

**Table 4 ijms-25-05074-t004:** A table containing the individual experiments of the articles selected according to the inclusion criteria for the analysis of the apoptosis results in continuous exposure mode. Results: non-significant (0); decrease (1); increase (2). Increasing order by frequency.

ID	Authors	Frequency (Hz)	Intensity (mT)	Exposure time (h)	Cell Type (T/NT)	Results	ID	Authors	Frequency (Hz)	Intensity (mT)	Exposure Time (h)	Cell Type (T/NT)	Results
1	Nezamtaheri, M. et al. (2022) [74]	0.01	1	2	T	2	35	Ledda, M. et al. (2018) [76]	10	1	2	T	0
2	Nezamtaheri, M. et al. (2022) [74]	0.01	1	2	NT	2	36	Ledda, M. et al. (2018) [76]	10	1	2	NT	0
3	Nezamtaheri, M. et al. (2022) [74]	0.01	1	2	T	0	37	Ledda, M. et al. (2018) [76]	10	1	2	T	0
4	Nezamtaheri, M. et al. (2022) [74]	0.01	1	2	T	0	38	Ledda, M. et al. (2018) [76]	10	1	2	T	0
5	Nezamtaheri, M. et al. (2022) [74]	0.01	1	24	T	0	39	Ledda, M. et al. (2018) [76]	10	10	2	T	0
6	Nezamtaheri, M. et al. (2022) [74]	0.01	1	24	NT	2	40	Ledda, M. et al. (2018) [76]	10	10	2	NT	0
7	Nezamtaheri, M. et al. (2022) [74]	0.01	1	24	T	0	41	Ledda, M. et al. (2018) [76]	10	10	2	T	0
8	Nezamtaheri, M. et al. (2022) [74]	0.01	1	48	T	0	42	Ledda, M. et al. (2018) [76]	10	10	2	T	0
9	Nezamtaheri, M. et al. (2022) [74]	0.01	1	48	NT	2	43	Ledda, M. et al. (2018) [76]	10	100	2	T	0
10	Nezamtaheri, M. et al. (2022) [74]	0.01	1	48	T	0	44	Ledda, M. et al. (2018) [76]	10	100	2	NT	0
11	Nezamtaheri, M. et al. (2022) [74]	0.01	1	72	T	0	45	Ledda, M. et al. (2018) [76]	10	100	2	T	0
12	Nezamtaheri, M. et al. (2022) [74]	0.01	1	72	NT	0	46	Ledda, M. et al. (2018) [76]	10	100	2	T	0
13	Nezamtaheri, M. et al. (2022) [74]	0.01	1	72	T	0	47	Ruiz Gómez, M.J. et al. (2001) [121]	25	1.5	2.75	T	0
14	Nezamtaheri, M. et al. (2022) [74]	0.01	1	120	NT	2	48	Ruiz Gómez, M.J. et al. (2001) [121]	25	1.5	2.75	T	0
15	Nezamtaheri, M. et al. (2022) [74]	0.01	100	2	T	0	49	Falone, S. et al. (2007) [94]	50	1	24	T	0
16	Nezamtaheri, M. et al. (2022) [74]	0.01	100	2	NT	0	50	Falone, S. et al. (2007) [94]	50	1	48	T	0
17	Nezamtaheri, M. et al. (2022) [74]	0.01	100	2	T	0	51	Falone, S. et al. (2007) [94]	50	1	72	T	0
18	Nezamtaheri, M. et al. (2022) [74]	0.01	100	2	T	0	52	Falone, S. et al. (2007) [94]	50	1	96	T	0
19	Nezamtaheri, M. et al. (2022) [74]	0.1	1	2	T	0	53	Garip, A. et al. (2010) [90]	50	1	3	T	1
20	Nezamtaheri, M. et al. (2022) [74]	0.1	1	2	NT	0	54	Ding, Z. et al. (2017) [129]	50	2.3	16	NT	0
21	Nezamtaheri, M. et al. (2022) [74]	0.1	10	2	T	0	55	Ding, Z. et al. (2017) [129]	50	2.3	16	NT	2
22	Nezamtaheri, M. et al. (2022) [74]	0.1	100	2	T	0	56	Yin, C. et al. (2016) [80]	50	8	90	NT	2
23	Nezamtaheri, M. et al. (2022) [74]	1	1	2	T	0	57	Ramazi, S. et al. (2023) [130]	50	20	24	T	0
24	Nezamtaheri, M. et al. (2022) [74]	1	10	2	T	0	58	Akbarnejad, Z. et al. (2017) [95]	100	10	72	T	0
25	Nezamtaheri, M. et al. (2022) [74]	1	100	2	T	0	59	Akbarnejad, Z. et al. (2017) [95]	100	10	72	T	0
26	Nezamtaheri, M. et al. (2022) [74]	1	100	2	NT	0	60	Akbarnejad, Z. et al. (2017) [95]	100	10	72	T	0
27	Nezamtaheri, M. et al. (2022) [74]	1	100	2	T	2	61	Akbarnejad, Z. et al. (2017) [95]	100	10	96	T	0
28	Nezamtaheri, M. et al. (2022) [74]	1	100	2	T	0	62	Akbarnejad, Z. et al. (2017) [95]	100	10	96	T	0
29	Nezamtaheri, M. et al. (2022) [74]	1	100	24	T	0	63	Akbarnejad, Z. et al. (2017) [95]	100	10	96	T	0
30	Nezamtaheri, M. et al. (2022) [74]	1	100	48	T	0	64	Akbarnejad, Z. et al. (2017) [95]	100	10	120	T	0
31	Nezamtaheri, M. et al. (2022) [74]	1	100	72	T	0	65	Akbarnejad, Z. et al. (2017) [95]	100	10	120	T	0
32	Nezamtaheri, M. et al. (2022) [74]	1	100	120	T	2	66	Akbarnejad, Z. et al. (2017) [95]	100	10	144	T	2
33	Ledda, M. et al. (2018) [76]	7	0.0125	120	NT	0	67	Akbarnejad, Z. et al. (2017) [95]	100	10	144	T	2
34	Ledda, M. et al. (2018) [76]	7	0.0125	120	NT	2							

**Table 5 ijms-25-05074-t005:** A table containing the individual experiments of the articles selected according to the inclusion criteria for the analysis of the oxidative stress results in continuous exposure mode. Results: non-significant (0); decrease (1); increase (2). Increasing order by frequency.

ID	Authors	Frequency (Hz)	Intensity (mT)	Exposure Time (h)	Cell Type (T/NT)	Results	ID	Authors	Frequency (Hz)	Intensity (mT)	Exposure Time (h)	Cell Type (T/NT)	Results
1	Nezamtaheri, M. et al. (2022) [74]	0.01	1	120	T	2	33	Iorio, R. et al. (2010) [131]	50	5	1	NT	0
2	Nezamtaheri, M. et al. (2022) [74]	0.01	1	120	NT	2	34	Iorio, R. et al. (2010) [131]	50	5	2	NT	2
3	Nezamtaheri, M. et al. (2022) [74]	0.01	1	120	T	0	35	Iorio, R. et al. (2010) [131]	50	5	3	NT	2
4	Nezamtaheri, M. et al. (2022) [74]	1	0.1	120	T	2	36	Patruno, A. et al. (2010) [132]	50	1	3	NT	2
5	Srdjenovic, B. et al. (2014) [125]	50	0.04	3	T	0	37	Patruno, A. et al. (2010) [132]	50	1	18	NT	2
6	Srdjenovic, B. et al. (2014) [125]	50	0.04	24	T	0	38	Wolf, F. et al. (2005) [120]	50	1	3	NT	0
7	Destefanis, M. et al. (2015) [123]	50	0.045	72	T	2	39	Wolf, F. et al. (2005) [120]	50	1	24	NT	0
8	Destefanis, M. et al. (2015) [123]	50	0.045	72	T	2	40	Morabito, C. et al. (2010) [92]	50	1	0.5	T	2
9	Destefanis, M. et al. (2015) [123]	50	0.045	168	T	2	41	Reale, M. et al. (2014) [93]	50	1	3	T	2
10	Destefanis, M. et al. (2015) [123]	50	0.045	168	T	2	42	Reale, M. et al. (2014) [93]	50	1	6	T	2
11	Calcabrini, C. et al. (2017) [83]	50	0.05	1	NT	1	43	Reale, M. et al. (2014) [93]	50	1	24	T	2
12	Calcabrini, C. et al. (2017) [83]	50	0.05	2	NT	2	44	Frahm, J. et al. (2006) [133]	50	1	0.75	NT	2
13	Calcabrini, C. et al. (2017) [83]	50	0.05	4	NT	0	45	Falone, S. et al. (2007) [94]	50	1	24	T	0
14	Frahm, J. et al. (2006) [133]	50	0.05	0.75	NT	2	46	Falone, S. et al. (2007) [94]	50	1	48	T	0
15	Morabito, C. et al. (2010) [92]	50	0.1	0.5	T	0	47	Falone, S. et al. (2007) [94]	50	1	72	T	0
16	Frahm, J. et al. (2006) [133]	50	0.1	0.75	NT	2	48	Falone, S. et al. (2007) [94]	50	1	96	T	0
17	Srdjenovic, B. et al. (2014) [125]	50	0.1	3	T	0	49	Yin, C. et al. (2016) [80]	50	8	1.5	NT	2
18	Srdjenovic, B. et al. (2014) [125]	50	0.1	24	NT	0	50	Ramazi, S. et al. (2023) [130]	50	20	24	T	2
19	Calcabrini, C. et al. (2017) [83]	50	0.1	1	NT	2	51	Kim, S. et al. (2017) [134]	60	0.8	20	NT	2
20	Calcabrini, C. et al. (2017) [83]	50	0.1	2	NT	2	52	Falone, S. et al. (2007) [94]	60	6	0.5	T	0
21	Calcabrini, C. et al. (2017) [83]	50	0.1	4	NT	0	53	Song, K. et al. (2018) [89]	60	6	72	T	1
22	Calcabrini, C. et al. (2017) [83]	50	0.15	1	NT	0	54	Song, K. et al. (2018) [89]	60	6	72	NT	1
23	Calcabrini, C. et al. (2016) [83]	50	0.2	1	NT	0	55	Song, K. et al. (2018) [89]	60	6	168	T	1
24	Frahm, J. et al. (2006) [133]	50	0.5	0.75	NT	2	56	Song, K. et al. (2018) [89]	60	6	168	NT	1
25	Patruno, A. et al. (2020) [135]	50	1	1	T	2	57	Akbarnejad, Z. et al. (2017) [95]	100	10	72	T	0
26	Patruno, A. et al. (2020) [135]	50	1	6	T	1	58	Akbarnejad, Z. et al. (2017) [95]	100	10	72	T	0
27	Patruno, A. et al. (2020) [135]	50	1	24	T	2	59	Akbarnejad, Z. et al. (2017) [95]	100	10	96	T	2
28	Park, J. et al. (2013) [88]	50	1	1.5	NT	2	60	Akbarnejad, Z. et al. (2017) [95]	100	10	96	T	2
29	Park, J. et al. (2013) [88]	50	1	96	NT	2	61	Akbarnejad, Z. et al. (2017) [95]	100	10	120	T	2
30	Garip, A. et al. (2010) [90]	50	1	3	T	2	62	Akbarnejad, Z. et al. (2017) [95]	100	10	120	T	2
31	Buldak, R. et al. (2012) [136]	50	1	0.66	T	0	63	Akbarnejad, Z. et al. (2017) [95]	100	10	144	T	2
32	Ayse, I. et al. (2010) [109]	50	5	1	T	2	64	Akbarnejad, Z. et al. (2017) [95]	100	10	144	T	2

**Table 6 ijms-25-05074-t006:** A table containing the individual experiments of the articles selected according to the inclusion criteria for the analysis of the cell cycle results in continuous exposure mode. Results: non-significant (0); significant (1). Increasing order by frequency.

ID	Authors	Frequency (Hz)	Intensity (mT)	Exposure Time (h)	Cell Type (T/NT)	Results	ID	Authors	Frequency (Hz)	Intensity (mT)	Exposure Time (h)	Cell Type (T/NT)	Results
1	Nezamtaheri, M. et al. (2022) [74]	0.01	1	2	T	1	27	Falone, S. et al. (2007) [94]	50	1	48	T	0
2	Nezamtaheri, M. et al. (2022) [74]	0.01	1	120	T	1	28	Falone, S. et al. (2007) [94]	50	1	72	T	0
3	Nezamtaheri, M. et al. (2022) [74]	0.01	1	120	NT	1	29	Falone, S. et al. (2007) [94]	50	1	96	T	0
4	Nezamtaheri, M. et al. (2022) [74]	0.01	1	120	T	1	30	Song, K. et al. (2018) [89]	60	6	24	T	0
5	Nezamtaheri, M. et al. (2022) [74]	0.01	1	120	T	1	31	Song, K. et al. (2018) [89]	60	6	24	NT	0
6	Nezamtaheri, M. et al. (2022) [74]	1	100	120	T	1	32	Song, K. et al. (2018) [89]	60	6	48	T	0
7	Nezamtaheri, M. et al. (2022) [74]	1	100	120	T	1	33	Song, K. et al. (2018) [89]	60	6	48	NT	0
8	Ruiz Gómez, M.J. et al. (2001) [121]	25	1.5	2.75	T	0	34	Song, K. et al. (2018) [89]	60	6	72	T	0
9	Ruiz Gómez, M.J. et al. (2001) [121]	25	1.6	2.75	T	0	35	Song, K. et al. (2018) [89]	60	6	72	NT	0
10	Patruno, A. et al. (2015) [118]	50	1	1	NT	1	36	Song, K. et al. (2018) [89]	60	6	168	T	0
11	Patruno, A. et al. (2015) [118]	50	1	24	NT	1	37	Song, K. et al. (2018) [89]	60	6	168	NT	0
12	Oh, I. Et al. (2020) [119]	50	1	120	T	1	38	Cho, S. et al. (2014) [81]	60	0.8	24	NT	0
13	Oh, I. Et al. (2020) [119]	50	1	120	T	1	39	Huang, C. et al. (2014) [126]	60	1.5	24	NT	0
14	Yin, C. et al. (2016) [80]	50	8	1.5	NT	1	40	Huang, C. et al. (2014) [126]	60	1.5	24	NT	0
15	Wolf, F. et al. (2005) [120]	50	1	12	NT	1	41	Huang, C. et al. (2014) [126]	60	1.5	48	NT	0
16	Wolf, F. et al. (2005) [120]	50	1	24	NT	0	42	Huang, C. et al. (2014) [126]	60	1.5	72	NT	0
17	Wolf, F. et al. (2005) [120]	50	1	24	NT	1	43	Huang, C. et al. (2014) [126]	60	1.5	96	NT	0
18	Wolf, F. et al. (2005) [120]	50	1	72	NT	0	44	Huang, C. et al. (2014) [126]	60	1.5	120	NT	0
19	Huang, C. et al. (2014) [126]	50	1.5	24	NT	0	45	Huang, C. et al. (2014) [126]	60	1.5	144	NT	0
20	Huang, C. et al. (2014) [126]	50	1.5	48	NT	0	46	Huang, C. et al. (2014) [137]	60	1.5	24	NT	0
21	Huang, C. et al. (2014) [126]	50	1.5	72	NT	0	47	Huang, C. et al. (2014) [137]	60	1.5	48	NT	0
22	Huang, C. et al. (2014) [126]	50	1.5	96	NT	0	48	Huang, C. et al. (2014) [137]	60	1.5	72	NT	0
23	Huang, C. et al. (2014) [126]	50	1.5	120	NT	0	49	Huang, C. et al. (2014) [137]	60	1.5	96	NT	0
24	Huang, C. et al. (2014) [126]	50	1.5	144	NT	0	50	Huang, C. et al. (2014) [137]	60	1.5	120	NT	0
25	Ma, Q. et al. (2014) (42)	50	2	72	NT	0	51	Huang, C. et al. (2014) [137]	60	1.5	144	NT	1
26	Falone, S. et al. (2007) (46)	50	1	24	T	0	52	Mihai, C. et al. (2014) [138]	100	5.6	0.75	NT	1

## Data Availability

Data are contained within this article or the Appendix A.

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
