# Peer review of "The Cellular Response Is Determined by a Combination of Different ELF-EMF Exposure Parameters: A Scope Review"

_ijms, 2024, doi:10.3390/ijms25105074_

Round 1

Reviewer 1 Report

Comments and Suggestions for Authors

To the authors

The reviewer has major and minor comments.

Major comments:

In recent decades, the numerous studies analyzing the biological effects of exposure have used a wide range of frequencies, amplitudes, and durations of exposure as well as cell lines (tumor or non-tumor), stem cells, tissue cultures in plastic Petri dishes or using scaffolds with different biophysical characteristics. However, signal symmetry, rise time and the shape of the applied signal have also been shown to influence the cellular response, aspects that are often not taken into account because they require an adequate level of technical expertise to achieve.

The authors underline the importance of controlling exposure parameters such as frequency, intensity, exposure time, exposure mode and report that it is not clear which is the most decisive in the cellular response. For decades intensity was considered the determining parameter, as well as time, exposure mode or waveform, but over the years no dose/time effect correlation has been found.

So, they continue....”one of the main problems in carrying out cellular experiments of exposure to magnetic fields is that there is no objective measurement of the different parameters, and the measurement of the frequency, intensity and real waveform that the cell receives is imprecise and inaccurate”.

….and more... “this leads to contradictory results between different research groups that use the same cell line, the same intensity and frequency, and even the same waveform, but fail to mimic the methodologies for monitoring these parameters and maintain methodological rigour”.

The authors do not consider in this review the measurement and effect of the geomagnetic field which strongly influences electromagnetic exposure experiments, they do not discuss it and do not report the published studies which have examined it. This very important phenomenon has been widely analyzed, studied and described in numerous articles (See Lisi at al. and Ledda at al.), which also report the correct control of the experimental environment, also considering the role of temperature during the exposure protocols.

-Therefore, to have a complete and exhaustive vision and not to report incorrect and imprecise information, they will have to discuss and rephrase/explain extensively, also reporting the scientific evidence of the following sentence "there is no objective measurement of the different parameters".

....and they will also have to rephrase/explain the following statement in light of new scientific evidences, page. 2 lines 51 to 56......”the fact that different combinations of exposure parameters are used makes it difficult to draw conclusions about the effects that these magnetic fields might produce in the different cell models studied, there are many publications that detect the non-homogeneity in the choice of parameters as the main reason for the non-replicability of the experiments and that claim that the results are inconclusive and incomparable among the existing publications”.

Minor comments:

-Please italicize the words "in vitro" throughout the manuscript.

-Please note that in the legends figures 2, 4, 5, 7, 8, 9, 10, and 11, in the results and in the discussion sections of the manuscript, it is important to use capital letters (A, B, C etc.) when referring to the figures.

-Please make the font and size consistent throughout the manuscript… see page 26, lines 39 to 49.

-Please check the manuscript for English fluency and typos.

Comments on the Quality of English Language

Minor editing of English language required

Reviewer 2 Report

Comments and Suggestions for Authors

The manuscript reviews articles on the effect of electromagnetic fields on mammalian cells. The analysis should be improved taking the following into account for refining the classification.

-tumor cells/non-tumor cells: are all of them cell lines of a mixture of primary cells and cell lines.

-differentiation between cells in suspension and adherent cells is important.

-cells isolated from 3D cultures or tumors should not be mixed with data from conventional 2D culture.

-the grouping into below and higher than 50Hz should be explained in the respective section

-the authors do not mention, which assays were used.

-human and murine cells need to be evaluated separately.

Round 2

Reviewer 2 Report

Comments and Suggestions for Authors

Limitations were mentioned.